# Vertical profile of tropospheric ozone derived from synergetic retrieval using three different wavelength ranges, UV, IR, and Microwave: sensitivity study for satellite observation

Yasuko Kasai[1,2], Tomohiro O. Sato[1], Takao M. Sato[3,1], Hideo Sagawa[4], Katsuyuki Noguchi[5], Naoko Saitoh[6], Hitoshi Irie[6], Kazuyuki Kita[7], Mona E. Mahani[1,8], Koji Zettsu[1], Ryoichi Imasu[9], and Sachiko Hayashida[5]

[1]National Institute of Information and Communications Technology, Tokyo, Japan
[2]Tokyo Institute of Technology, Tokyo, Japan
[3]Institute of Space and Astronautical Science, Japan Aerospace Exploration Agency, Kanagawa, Japan
[4]Kyoto Sangyo University, Kyoto, Japan
[5]Nara Women's University, Nara, Japan
[6]Center for Environmental Remote Sensing, Chiba University, Chiba, Japan
[7]Ibaraki University, Ibaraki, Japan
[8]Tohoku University, Sendai, Japan
[9]Atmospheric and Ocean Research Institute, The University of Tokyo, Chiba, Japan

*Correspondence to:* Y. Kasai (ykasai@nict.go.jp)

**Abstract.** We performed a feasibility study to constraining the vertical profile of the amount of ozone in the troposphere by using a synergetic retrieval method on multiple spectra, i.e., ultraviolet (UV), thermal infrared (TIR) and microwave (MW) ranges, measured from space. A quantitative evaluation of the sensitivity of the tropospheric retrieval by adding the MW measurement to the UV and TIR measurements was reported for the first time by this work. The urban and sea areas in East Asia in summer and winter seasons were selected for the feasibility study. Geometry of line-of-sight was nadir down-looking for UV and TIR measurements, and limb-sounding for MW measurement. The sensitivities of retrieved ozone in the upper troposphere (UT), middle troposphere (MT) and lowermost troposphere (LMT) were estimated using values of the degree of freedom for signal (DFS), pressure of maximum sensitivity, error reduction rate from the a priori error, and averaging kernel matrix, derived based on the optimal estimation method. The measurement noises were assumed at the same level as the currently available instruments. The weighting functions for the UV, TIR and MW ranges were calculated using the SCIATRAN radiative transfer model, the Line-By-Line Radiative Transfer Model, and the Advanced Model for Atmospheric Terahertz Radiation Analysis and Simulation, respectively. The DFS value was increased by approximately 96 %, 23 % and 30 % by adding the MW measurements to the combination of UV and TIR measurements in the UT, MT and LMT regions, respectively. We found that the MW measurement increased the DFS value of the LMT ozone; nevertheless, the MW measurement alone has no sensitivity for the LMT ozone. The pressure of maximum sensitivity of the LMT ozone was also increased by adding the MW measurement. It might indicate that better information of the LMT ozone can be educed by adding constraints on the UT and MT ozone from the MW measurement. The results of this study will be implemented in the Japanese air-quality monitoring missions, APOLLO, GMAP-Asia and uvSCOPE.

# 1 Introduction

The World Health Organization (WHO) estimates that around seven million people died as a result of the effects of air pollution in 2012 (WHO, 2014), and it cites air pollution as being one of the world's largest single environmental health risk. Ozone in particular causes serious damage for human health and agricultural crops. Tropospheric ozone has been increasing globally at rates of 0.3–1.0 ppb yr$^{-1}$ over past few decades in the northern hemisphere (Dentener et al., 2010, and references therein). Ozone is formed by sunlight-driven oxidation from ozone precursors such as methane ($CH_4$), carbon monoxide (CO), non-methane volatile organic compounds (NMVOCs), and nitrogen oxides ($NO_x$) in the troposphere. Monitoring of the amount of the tropospheric ozone is required to understand the current status and to make forecasts of future ozone amount.

Ozone plays different roles in different altitude regions in the troposphere. It is well known that surface ozone is a harmful pollutant that has a detrimental impact on the health of humans and plants and is responsible for significant reduction in crop yields. The lifetime of ozone in the free troposphere ranges from a few days to weeks, so that the transport scale of ozone is potentially intercontinental and hemispheric. Upper tropospheric ozone is the third most important warming gas and is responsible for a large part of the human enhancement of the global greenhouse effect. For further understanding these different characteristics of tropospheric ozone, it is important to obtain information on the vertical distribution of ozone separately in the lowermost troposphere (LMT), middle troposphere (MT), and upper troposphere (UT) on a global scale.

Ozone has been observed from space in a variety of spectral ranges, including the ultraviolet (UV), visible (VIS), thermal infrared (TIR), and microwave (MW) with different observation geometries (nadir-looking and limb-sounding). Observations at different wavelengths have sensitivity to ozone at different altitudes. Generally, nadir-looking observations in the UV/VIS range are sensitive to ozone in the LMT (e.g., the Ozone Monitoring Instrument, OMI, onboard the Aura satellite (Levelt et al., 2006) and the second Global Ozone Monitoring Experiment, GOME-2, onboard the MetOp satelle (Munro et al., 2006)), while nadir-looking in the TIR range is sensitive to ozone in the MT (e.g., the Thermal Emission Spectrometer, TES, onboard the Aura satellite (Osterman et al., 2008) and the Infrared Atmospheric Sounding Interferometer, IASI, onboard the MetOp satellites (Clerbaux et al., 2009)). Limb-sounding and stellar/solar-occultation is used to sound ozone in the stratosphere and above. Limb-sounding in the UV/VIS region sounds ozone in the stratosphere, and stellar occultation instruments observe ozone above the stratosphere (e.g., the Scanning Imaging Absorption Spectrometer for Atmospheric Chartography, SCIAMACHY, (Brinksma et al., 2006) and the Global Ozone Monitoring by Occultation of Stars, GOMOS, (Kyrölä et al., 2004) both onboard the Envisat satellite). Limb-sounding in the MW spectral range is sensitive at altitudes above the UT (e.g., the Microwave Limb Sounder, MLS, onboard the Aura satellite (Waters et al., 2006) and the Superconducting Submillimeter-Wave Limb-Emission Sounder, SMILES, onboard the International Space Station (Kikuchi et al., 2010)).

Measurement using several wavelength ranges is an advanced method of deriving a vertically resolved ozone profile. Ziemke et al. (2006) derived the global distribution of the tropospheric ozone column by subtracting the stratospheric ozone column measured using the MLS MW spectra from the total ozone column measured using the OMI UV spectra. A feasibility study of the tropospheric ozone retrieval using the optimal estimation method (OEM) (Rodgers, 2000) combining UV and TIR measurements was performed by Landgraf and Hasekamp (2007). Worden et al. (2007) implemented the concept of synergetic

retrieval on the OMI and TES measurements. Natraj et al. (2011) showed that the retrieval sensitivity of the LMT is improved by combining UV and TIR measurements. Fu et al. (2013) implemented a synergetic retrieval of boundary layer ozone using the UV and TIR spectra of the OMI and TES measurements. A value of the degree of freedom for signal (DFS) for ozone from the surface to 700 hPa was estimated to be 0.37±0.09 for 22 coincident measurements among OMI, TES, and ozonesonde

from 2004 to 2008 (see Table 2 in Fu et al. (2013)). Cuesta et al. (2013) also performed a synergetic retrieval of boundary layer ozone, using the GOME-2 (for UV) and IASI (for TIR) measurements. The DFS values for ozone up to 3 km were estimated to be 0.34±0.04 and 0.23±0.04 over land and ocean, respectively, on 19–20 August 2009 over Europe (see Table 1 in Cuesta et al. (2013)). The corresponding heights of maximum sensitivity were 2.20±0.50 km and 3.42±0.59 km, respectively. The DFS values were approximately 50 % increased by combining the GOME-2 and IASI measurements, compared with the IASI

measurement only. The other approach to retrieve the tropospheric ozone profile using neural network technique was performed with the SCIAMACHY nadir measurements in the UV and VIS ranges (Sellitto et al., 2012a, b). They also showed a significant availability of combining several wavelength ranges to retrieve the tropospheric ozone profile.

Our idea is to add MW measurements to the synergetic retrieval of the tropospheric ozone. To the best of our knowledge, no study has attempted to show how MW measurements improve the retrieval of the vertical profile of tropospheric ozone.

In this study, we performed a feasibility study of obtaining a vertically resolved ozone amount in the troposphere by using synergetic retrieval from a combination of UV, TIR and MW measurements covering wide wavelength ranges. This work can be of benefit to future missions for air-pollution, such as Air POLLution Observation (APOLLO), Geostationary mission for Meteorology and Air Pollution (GMAP-Asia) (Kasai et al., 2011), uvSCOPE (Fujinawa et al., 2015), and air pollution prediction project in National Institute of Information and Communication Technology (NICT). The objective of APOLLO

and GMAP-Asia mission is to measure short-lived climate pollutants for monitoring global pollution and climate change. The missions of APOLLO and GMAP-Asia assume atmospheric monitoring from the International Space Station (ISS) and from geostationary orbit, respectively. The uvSCOPE mission, a candidate for the earth observation section of ISS, aims to detect hot-spots of air pollutant with a high horizontal resolution (such as $1\times1\,km^2$) for better understanding of the inventory of air pollution. The target of NICT air pollution prediction project is to make health index to mitigation of air pollution disasters

using high-horizontal resolution (a few km scale) pollution forecasting from multiple data-sets, such as satellite observation, ground-based observation, and in-situ observation data.

In this paper, we report a feasibility study of the tropospheric ozone retrieval based on the concept of APOLLO, i.e., to obtain vertically resolved information of ozone within the troposphere not only at the boundary layer but also in the middle and upper troposphere by utilizing synergetic observation afforded by UV, TIR and MW instruments. Our major aim of this

feasibility study is to evaluate the sensitivity of the tropospheric ozone retrieval adding the MW measurement to the multi-spectral synergetic retrieval. Thus, the feasibility study was performed under an ideal condition for the synergetic retrieval of the tropospheric ozone.

## 2 Observation scenario

### 2.1 Observation wavelength region and geometry

The observation scenario follows the concept of the APOLLO mission. We assumed three spectrometers equipped in ISS that observe the three wavelength ranges of UV, TIR, and MW. Figure 1 shows the observation geometries for the three spectrometers. The UV and TIR instruments use nadir down-looking, and the MW measurement uses limb-sounding at tangent heights from 10 to 80 km. In this feasibility study, we assumed spherically homogeneous atmosphere along the line-of-sight of the MW measurement. The height of ISS was assumed to be 300 km in our simulation. The azimuthal direction of the field-of-view of the MW limb-sounding was set to parallel to the ISS's orbital motion. The tangent point of the MW limb-sounding passes the UV and TIR nadir down-looking point approximately five minutes before the UV and TIR nadir down-looking. The time delay of 5 minutes corresponds to approximately 6 km transport when we assume a typical value of horizontal wind speed in the troposphere and the stratosphere of 1.2 km/min (Fleming et al., 1988), which is smaller than typical horizontal resolution of the MW limb measurement (8 km for Aura/MLS (Waters et al., 2006)). Therefore, we ignored this time difference between the UV and TIR measurements and MW measurement. Table 1 is a summary of the specification of the three assumed instruments and the radiative transfer models used in this study.

The wavelength range of the simulation of the UV measurement was set to 305–340 nm. The UV wavelength ranges shorter than 305 nm is useful for the stratospheric ozone retrieval (e.g., Bak et al., , 2012). In our simulation, we added the MW limb measurement which is more sensitive for the stratospheric ozone. We excluded the shorter UV wavelength ranges to clearly show whether the stratospheric ozone retrieval by the MW measurement improves the tropospheric ozone retrieval sensitivity. We also decided not to include the VIS (340–505 nm) range in this study, although the benefit of adding VIS wavelengths has been reported (e.g., Sellitto et al., 2012a, b). The reason why we excluded these ranges is because the wavelength dependence of the surface reflectance, absorption of $NO_2$ and the Ring effect were out of the scope of the study. The spectral resolution, defined as the full width at half maximum (FWHM), and the sampling step were assumed to be 0.6 nm and 0.2 nm, respectively. The noise equivalent spectral radiance (NESR) was obtained by dividing the simulated backscattered radiance by the signal-to-noise ratio (SNR). We used SNR values of the APOLLO instrument setups in the UV simulation. Three references of the SNR value were prepared for high, middle and low level of the radiance. The SNR value for the simulated radiance was linearly interpolated by the two of three reference SNR values. Table 2 shows the reference SNR values at 305 nm and 340 nm, and the solar zenith angle (SZA) and surface albedo of the three conditions [Private communication with K. Gerilowski]. The values of SNR used in the UV simulation were estimated to be approximately 90 and 1400 at 305 nm and 340 nm, respectively.

We assumed that the nadir-viewing TIR instrument would be a Fourier transform spectrometer covering the TIR spectral range (980–1080 $cm^{-1}$) including the ozone v3 absorption band, 9.6 $\mu$m (1045 $cm^{-1}$) as TES (Osterman et al., 2008) and IASI (Clerbaux et al., 2009). We set the maximum optical path difference to 8.33 cm, which corresponds to a spectral resolution of 0.12 $cm^{-1}$ and calculated the noise equivalent differential temperature for each wavelength, assuming that the SNR is a constant value of 300 in the entire spectral range.

Several ozone transitions in the microwave/submillimeter range have been employed by recent space-borne instruments, e.g., 206.1 and 235.7 GHz for Aura/MLS (Waters et al., 2006), 501.8 and 544.6 GHz for Odin/SMR (Urban et al., 2005), and 625.4 GHz for Aura/MLS and JEM/SMILES (Kikuchi et al., 2010). The MW limb-sounding instrument considered in this study was designed for covering two frequency bands, i.e., the 350 GHz band (345–357 GHz) and 645 GHz band (639–651 GHz). There are ozone lines at 352.3, 352.8, and 355.0 GHz in the former and at 640.1, 642.3, 644.8, 645.6, 647.8, and 650.7 GHz in the latter. These frequency bands were selected for detection of not only ozone but also other molecules related to global warming and air-pollution ($H_2O$, CO, $CH_3CN$, $N_2O$, $SO_2$, $H_2CO$, and $HNO_3$). The channel separation width of the spectrometer was assumed to be 25 MHz. The frequency resolution, defined by FWHM was set to be identical to the channel separation width. The antenna diameter was assumed to be 40 cm. The Earth's limb was assumed to be scanned vertically from 10 to 80 km with an interval of 2 km and total 35 spectra were acquired in one vertical limb scan. We also assumed that a typical integration time was 0.5 seconds for one spectrum accumulation and it took 17.5 seconds for one vertical limb scan. The brightness temperature noise was estimated to be 0.7 K and 1.7 K for the 350 GHz and 645 GHz bands, respectively, assuming the system noise temperature of a typical Shottky-barrier mixer (2500 K and 6000 K for 350 GHz band and 645 GHz band, respectively).

## 2.2 Atmospheric conditions

We performed a feasibility study of tropospheric ozone retrieval for typical atmospheric scenarios in summer and winter. The target area of this study is East Asia, one of most serious ozone polluted areas and the source of ozone intercontinental transport toward North America from Asia. We chose two observation points in East Asia, 35°N, 116.5°E (Central-East China, CEC, located between Beijing and Shanghai) and 31°N, 127.25°E (East China Sea, ECS). The CEC is the area where largest amount of the boundary layer ozone was observed from the Aura/OMI measurement (Hayashida et al., 2015). The ECS was chosen for a comparison of urban area and ocean. We selected June and December for representatives of summer and winter seasons, respectively. Hayashida et al. (2015) reported the amount of the boundary layer ozone in the area near CEC was maximized in June and minimized in December. The observation time was set to 04:00am GMT, which corresponds to 11:46am and 00:29pm for CEC and ECS local times, respectively. The local times around noon were set because it was shown that the ozone retrieval sensitivity was most increased in small to moderate SZAs in the simulation study performed by Landgraf and Hasekamp (2007).

We made a total of 20 atmospheric scenarios over the two Asian areas (CEC and ECS) in June and December 2009. The characteristics of the 20 atmospheric scenarios are presented in Table 3, and the vertical profiles of ozone, temperature and water vapor are shown in Figure 2. We discuss the simulation results, dividing into four cases (CEC in June, ECS in June, CEC in December and ECS in December). The ozone partial column (PC) in LMT at CEC in June was largest (approximately $5 \times 10^{21}$ $m^{-2}$) among those of four cases, and the smallest value (approximately $2 \times 10^{21}$ $m^{-2}$) was taken from the case at CEC in December.

We interpolated the values from the following three original atmospheric profiles by using cubic splines to make the atmospheric profile smooth in the overlapping regions for a vertical pressure ($p$) grid defined as follows.

$$p[i] = \begin{cases} 10^{3-(i+1)/24}\,[\text{hPa}] & i = 1, 2, \ldots, 71 \; (\geq 1\,\text{hPa}) \\ 10^{3-(i-35)/12}\,[\text{hPa}] & i = 72, 73, \ldots, 108 \; (< 1\,\text{hPa}) \end{cases}$$

The scale height of the vertical profiles that we used was 3 km.

The profiles of ozone, temperature, and water vapor in the vertical region from the surface to 65 hPa (approximately 20 km) were simulated by a one-way nested global-regional air quality forecasting (AQF) system (Takigawa et al., 2007, 2009). This system is based on the CHASER (Chemical Atmospheric General Circulation model for the Study of Atmospheric Environment and Radiative Forcing) model (Sudo et al., 2002) and WRF (Weather Research and Forecasting)/Chem model (Grell et al., 2005) version 3.3. The horizontal resolution of this system is approximately 40 km. The profiles over CEC and ECS were
spatially averaged for the periods of June 1 to June 30 and December 1 to December 31, 2009. The surface temperature was simulated with the AQF system, and the temperature difference between the surface and the lower boundary of the lowest atmospheric layer was less than 1 K. We set the surface temperature to be equal to the value at the surface pressure, since the effect of the temperature contrast between the atmosphere and surface is large for the TIR measurement.

The profiles (ozone, temperature and water vapor) in a vertical region of 985–0.01 hPa were taken from the Modern
Era Retrospective-Analysis for Research and Applications (MERRA) data (Rienecker et al., 2011). A data product named "MERRA DAS 3d analyzed state (inst6_3d_ana_Nv)" provided the three-dimensional fields of layer pressure thickness, air temperature, specific humidity, and ozone mixing ratio at six-hour intervals (00:00, 06:00, 12:00, 18:00 GMT). The MERRA data covered a $0.66° \times 0.5°$ latitude-longitude grid.We averaged the MERRA data at 06:00 GMT (the nearest local time of 12:00 LT in CEC and ECS) on the same date of the selected AQF system profiles for each region (CEC and ECS). No
interpolation for local time was performed on the MERRA data.

The temperature data of the COSPAR International Reference Atmosphere (CIRA) (Fleming et al., 1990) was used above the vertical level of 0.01 hPa. The CIRA-86 includes monthly and zonally mean temperatures and pressures (0–120 km) with almost global coverage (80°N–80°S) at an interval of 10°. We averaged the two temperature data at 30°N and 40°N for CEC, and used the temperature data at 30°N for ECS. The mixing ratios of ozone and water vapor at pressures less than 0.01 hPa
were assumed to be equal to those at the upper boundary (0.01 hPa) of the MERRA data because there are no appropriate data to refer. We confirmed that the effects of the assumption in the upper vertical range were negligibly small for our calculation.

We assumed the following quasi-clear sky cases for all scenarios. A no-cloud condition was considered for all wavelength ranges. Basic background aerosol was taken into account only in the UV calculation. The aerosol profile was included as a known parameter because it was reported that the inclusion of aerosol profile changed within 2 % of the LMT ozone amount
in case of the Aura/OMI measurement (Hayashida et al., 2015). We used the vertical profiles of urban and maritime aerosols of a standard mixing state that were described in Hess et al. (1998). These profiles were adjusted to be 0.2 of the total optical thicknesses of the aerosols (moderate pollution). The aerosol profile was not included in the TIR calculation. The extinction of radiation due to aerosol particles with a scale of approximately 9.6 $\mu$m, which corresponds to the wavelength of the TIR

range, is negligibly small for the synergetic retrieval of the LMT ozone with the TIR measurement (e.g., Natraj et al., 2011). We assumed that surface albedo was constant in the selected UV ranges (305–340 nm). The information on surface albedo for simulating UV radiance spectra was taken from the database described by Kleipool et al. (2008). This database contains the monthly global maps of the Earth's surface Lambertian equivalent reflectance (LER) deduced from the Aura/OMI measure-

ments. We obtained monthly and spatially averaged albedo values of 0.056 (June) and 0.063 (December) for CEC and 0.065 (June) and 0.084 (December) for ECS, respectively, from the LER data at the wavelength of 328.1 nm, which is the shortest wavelength in the database. The impact of the uncertainty of the UV surface albedo on tropospheric ozone measurements from space was discussed in Noguchi et al. (2014). The surface emissivity for modeling the TIR radiance spectra was estimated by linear regression analysis based on the Advanced Space-borne Thermal Emission Reflection Radiometer (ASTER) Spectral

Library (Baldridge et al., 2009). The surface emissivity for MW was set to 1.0 for the entire range. MW limb measurements are generally insensitive to the surface emissivity since the atmosphere is strongly opaque in this wavelength range.

## 3    Synergetic retrieval simulation

### 3.1    Forward models of UV, TIR, and MW regions

We used the SCIATRAN radiative transfer model version 3.1 (Rozanov et al., 2005), the Line-By-Line Radiative Transfer

Model (LBLRTM) version 12.1 (Clough et al., 2005), and the Advanced Model for Atmospheric Terahertz Radiation Analysis and Simulation (AMATERASU) (Baron et al., 2008) for the calculation of spectra in the UV, TIR, and MW wavelength ranges, respectively. In the presented study, no bias is assumed between the three forward models in order to investigate potential advantage of including MW observation to retrieval of tropospheric ozone.

The SCIATRAN model was developed by the Institute of Remote Sensing/Institute of Environmental Physics (IFE/IUP)

of the University of Bremen, Germany, for fast and precise simulation of radiance spectra in the UV, VIS and Near Infrared ranges as measured by spaceborne instruments, e.g., GOME (240–790 nm) and SCIAMACHY (240–2400 nm). SCIATRAN is applicable to spectral regions ranging from 175.44 nm to 2400 nm, and is basically compatible with arbitrary observation geometries and sensor positions in space, in the atmosphere, and on the ground. The spherical shape of the Earth's atmosphere, including the refraction effect, is properly taken into account when simulating the radiance spectra.

The LBLRTM model is an accurate and efficient line-by-line radiative transfer model, and it has been extensively validated for atmospheric radiance spectra from UV to submillimeter-wave ranges. The line-by-line calculation of the optical thickness of the atmospheric layers is conducted on the basis of the spectroscopic line parameter database (HITRAN 2008) with its updates (Rothman et al., 2009). This model is used as the forward model in retrieval algorithms for analyzing spaceborne measurements such as EOS-Aura/TES (Clough et al., 2006), and GOSAT/TANSO-FTS (Saitoh et al., 2009).

The AMATERASU model consists of a line-by-line radiative transfer calculation allowing for a multi-layered horizontally homogeneous shell atmosphere. This model has been implemented in the retrieval analysis of the SMILES measurements (e.g., Baron et al., 2011) and in the feasibility study of a submillimeter instrument for planetary science (Kasai et al., 2012). The spectroscopic parameters are from commonly used databases such as HITRAN 2008 (Rothman et al., 2009) and the JPL

spectroscopic catalog (Pickett et al., 1998). The continuum absorption due to dry and wet air are also included and are based on the formulation in Pardo et al. (2001).

## 3.2 Theoretical retrieval basis and error estimation

The optimal estimation method (OEM) (Rodgers, 2000) was used for the synergetic retrieval system and their error estimations. The retrieved state vector $\hat{x}$ was estimated by minimizing the differences between the observed radiance spectra $y_{\text{obs}}$ and the modeled radiance spectra $y_{\text{mod}}$, using a constraint from an a priori state vector $x_{\text{a}}$.

$$\hat{x} = Ax + (I - A)\,x_{\text{a}} + G\epsilon \tag{1}$$

In this equation, $x$ is the true state vector, $A$ is the averaging kernel matrix, $G$ is the gain (contribution function) matrix, and $\epsilon$ is the measurement noise vector. The averaging kernel matrix characterizing the sensitivity of the retrieved state vector $\hat{x}$ to the true state vector $x$ is given by

$$A = \frac{\partial \hat{x}}{\partial x} = GK = \left(K^T S_\epsilon^{-1} K + S_{\text{a}}^{-1}\right)^{-1} K^T S_\epsilon^{-1} K, \tag{2}$$

where $S_{\text{a}}$ and $S_\epsilon$ are the a priori covariance matrix and the measurement error covariance matrix, respectively. $K$ is a weighting function matrix ($K = \partial y_{\text{mod}}/\partial x$). $A$ corresponds to the identity matrix when the retrieved profile is equal to the true atmospheric profile. The number of state vector elements which are independently resolved is obtained by summation of diagonal elements of $A$, and is defined as DFS. The $i$th element of measurement response vector, $m[i]$, is defined as

$$m[i] = \sum_j A[i,j]. \tag{3}$$

A value of the measurement response element near unity indicates that almost all information in the retrieval result comes from the observation spectra, while a small value indicates that the retrieval result is largely influenced by the a priori.

The total retrieval error covariance $\hat{S}$ is calculated using the covariance matrices of the smoothing error $S_{\text{s}}$ and measurement noise $S_{\text{m}}$.

$$\begin{aligned}
\hat{S} \ &= S_{\text{s}} + S_{\text{m}} \\
&= (I - A)\,S_{\text{a}}\,(I - A)^T + GS_\epsilon G^T \\
&= \left(K^T S_\epsilon^{-1} K + S_{\text{a}}^{-1}\right)^{-1}
\end{aligned} \tag{4}$$

The square root of the $\hat{S}$ diagonals is the total retrieval error in $\hat{x}$ ($\epsilon_{x}$). The value of $\epsilon_{x}$ at $i$th layer is given by

$$\epsilon_{x}[i] = \sqrt{\hat{S}[i,i]}. \tag{5}$$

We evaluated the sensitivity of the vertical profile of ozone from the synergetic retrieval for seven different combinations of the wavelength ranges, i.e., UV, TIR, MW, TIR+MW, UV+MW, UV+TIR and UV+TIR+MW, in the 20 atmospheric scenarios. The state vectors $x$, $\hat{x}$ and $x_{\text{a}}$ were calculated using logarithm units of the volume mixing ratio (VMR). The diagonal

components of $S_a$ were the squares of the a priori error $\sigma_a$ at each vertical pressure grid). The value of $\sigma_a$ was set to $100\,\%$ of the log-based a priori VMR to simply quantify the error reduction from the error in the a priori error to the error in the retrieved state due to the measurement. The diagonal components of $S_\epsilon$ were the squares of the measurement error $\sigma_\epsilon$. The off-diagonal components of $S_\epsilon$ were set to zero. The off-diagonal components in $S_a$ indicate the correlations between the ozone concentrations in different vertical layers. Non-zero off-diagonal components in $S_a$ assist the retrieval of ozone concentration in a layer in which sufficient ozone information is not included in a measurement spectrum with the correlations with other layers in which sufficient ozone information is included in a measurement spectrum (e.g., Saitoh et al., 2009). One of the aims of our feasibility study is to investigate the ozone retrieval sensitivity for each vertical region in an ideal condition. We set off-diagonal components in $S_a$ to be zero to avoid the assistance of the ozone retrieval by the correlations between different vertical layers.

We normalized the state vector $\boldsymbol{x}$ and measurement vector $\boldsymbol{y}$ with $\sigma_a$ and $\sigma_\epsilon$ because values with different order in a vector and a matrix often cause undesirable mathematical errors in computational calculation.

$$\boldsymbol{u} \quad = \frac{\boldsymbol{x} - \boldsymbol{x}_a}{\sigma_a} \tag{6}$$

$$\boldsymbol{v} \quad = \frac{\boldsymbol{y}_{\mathrm{obs}} - \boldsymbol{y}_{\mathrm{mod}}}{\sigma_\epsilon} \tag{7}$$

The normalized weighting function was given by

$$K' = KD\left(\sigma_a/\sigma_\epsilon\right). \tag{8}$$

Here, $D\left(\boldsymbol{a}\right)$ is a diagonal matrix whose diagonal elements are equal to the components of the vector $\boldsymbol{a}$. $S_a$ and $S_\epsilon$ were normalized in the same way.

$$S'_a = D\left(1/\sigma_a\right) S_a D\left(1/\sigma_a\right)^T \tag{9}$$

$$S'_\epsilon = D\left(1/\sigma_\epsilon\right) S_\epsilon D\left(1/\sigma_\epsilon\right)^T \tag{10}$$

Using the normalized vectors and matrices, $A$ and $\hat{S}$ are expressed as

$$A = \left(K'^T S'^{-1}_\epsilon K' + S'^{-1}_a\right)^{-1} K'^T S'^{-1}_\epsilon K', \tag{11}$$

$$\hat{S} = \left(K'^T S'^{-1}_\epsilon K' + S'^{-1}_a\right)^{-1}. \tag{12}$$

We evaluated the sensitivity of ozone retrieval for seven wavelength combinations in terms of DFS. We calculated the DFS values for the partial column in the UT, MT and LMT regions. The value of DFS from the $i_{\mathrm{min}}$th vertical layer to the $i_{\mathrm{max}}$th layer is given by

$$\mathrm{DFS} = \sum_{i=i_{\mathrm{min}}}^{i_{\mathrm{max}}} A[i,i]. \tag{13}$$

We also evaluated the sensitivity of ozone retrieval using the pressure of maximum sensitivity (PMS) and the reduction rate of error (RRE) for the partial column. The PMS was defined as the pressure of the maximum of the sum of rows of the

corresponding $A$ for the ozone partial column. The RRE is given by

$$\text{RRE} = \frac{\text{PCE}_{\text{apriori}} - \text{PCE}_{\text{retrieved}}}{\text{PCE}_{\text{apriori}}} \quad [\%] \tag{14}$$

where $\text{PCE}_{\text{apriori}}$ and $\text{PCE}_{\text{retrieved}}$ are the partial column error, PCE, for the a priori state and the retrieved state, respectively. PC represents the partial column of ozone, and the value of PC from the $i_{\text{min}}$th vertical layer to the $i_{\text{max}}$th layer is given by

$$\text{PC} = \sum_{i=i_{\text{min}}}^{i_{\text{max}}} \frac{p[i] \cdot \text{VMR}[i]}{k_{\text{B}} \cdot T[i]} \Delta z[i]. \tag{15}$$

Here, $p[i]$, $\text{VMR}[i]$, $T[i]$ and $\Delta z[i]$ are pressure, VMR of ozone, temperature, and the vertical length of the $i$th layer, respectively. $k_{\text{B}}$ is the Boltzmann constant. The PCE is given by

$$\text{PCE} = \sum_{i=i_{\text{min}}}^{i_{\text{max}}} \frac{p[i] \cdot \epsilon_{\text{VMR}}[i]}{k_{\text{B}} \cdot T[i]} \Delta z[i]. \tag{16}$$

$\epsilon_{\text{VMR}}[i]$ is the total retrieval error in ozone VMR at the $i$th layer ($\sigma_{\text{a}}$ for $\text{PCE}_{\text{apriori}}$ and $\epsilon_{\boldsymbol{x}}$ for $\text{PCE}_{\text{retrieved}}$).

## 4   Results and discussion

The sensitivity of ozone retrieval for the UT (215–383 hPa), MT (383–749 hPa), and LMT (>749 hPa) regions was investigated in terms of DFS. Figure 3 and Table 4 show the DFS values calculated with Eq. (13) for the seven wavelength combinations: UV alone, TIR alone, MW alone, TIR+MW, UV+MW, UV+TIR and UV+TIR+MW. The DFS values were averaged in June in CEC (shown by red markers in Fig. 3), June in ECS (purple), December in CEC (green), December in ECS (blue) and all 20 profiles (black). The error bar represents the standard deviation.

The DFS value in the UT region averaged for all 20 profiles was calculated to be 0.16±0.08, 0.59±0.10 and 0.44±0.41 for the UV, TIR, and MW wavelength range, respectively. None of the DFS average values for one wavelength range was larger than unity. Using more than one wavelength range, the DFS value increased to 1.15±0.25, 0.90±0.30, 0.62±0.08, and 1.21±0.28, for the wavelength combinations of TIR+MW, UV+MW, UV+TIR, and UV+TIR+MW, respectively. The DFS of the UV+TIR combination was the lowest among those of more than one wavelength range, and adding the MW region increased the value by about two times. The additional MW region was hence most effective at improving the retrieval of ozone in the UT region.

In the MT region, the TIR measurements are the main contributors of DFS information. The DFS values were 0.50±0.16, 0.83±0.11, and less than 0.01 for the UV, TIR and MW wavelength ranges, respectively. The DFS values increased in the same way as in the UT calculation by adding measurements in different wavelength ranges. The average DFS values of the 20 profiles were 1.03±0.09, 0.73±0.09, 1.00±0.09 and 1.23±0.13 for TIR+MW, UV+MW, UV+TIR and UV+TIR+MW, respectively. It should be noted that the MW measurements, which have no information on ozone in the MT region because of atmospheric opacity, certainly increased the DFS value in the MT region from 1.00 to 1.23 (about 23 % increase) for the TIR+UV measurements. This indicates that the information on ozone in the stratosphere and UT, where the sensitivity of MW is high, is also important for retrieval of ozone in the MT region.

The DFS values in the LMT region were generally smaller than those in the UT and MT regions. They were calculated to be 0.20±0.13, 0.21±0.15, less than 0.01, 0.20±0.14, 0.26±0.15, 0.46±0.25 and 0.60±0.27 for UV, TIR, MW, TIR+MW, UV+MW, UV+TIR and UV+TIR+MW, respectively. The DFS values of the UV and TIR wavelength ranges were almost the same, while the MW measurements had no sensitivity in the LMT region. Similar to the MT region, the DFS values of the UV+TIR measurements (0.46) increased to 0.60 (about 30 % increase) as a result of adding the MW measurement. We note that the DFS value for the TIR measurement in the case at ECS in December 2009 was larger than that for TIR+MW measurement. The averaging kernel matrix in the LMT region for the TIR measurement of this case was discontinuously large. In general, the discontinuity in the averaging kernel occurs because of mathematical issues not atmospheric physical issues, thus we avoid any scientific discussion with the DFS value for the TIR measurement in this case (ECS in December 2009).

We compared our results with the previous studies of estimating the tropospheric ozone sensitivity using DFS. The DFS values for UV, TIR and UV+TIR measurements were summarized in Table 5. Scenarios of the simulation or the measurements are different among this work and the previous studies shown in Table 5, thus, the DFS values themselves should not be directly compared each other. Here we calculated the relative difference between the DFS value for the UV+TIR measurements and mean of the DFS values for UV and TIR measurements. The relative difference of our simulation for all profiles averaged was estimated to be 126 %. It showed good agreement with those of Fu et al. (2013) (139 %), Cuesta et al. (2013) (104 %) and Natraj et al. (2011) (115 %).

The pressure of maximum sensitivity, PMS, for the ozone partial column should be located in a range of the corresponding partial column. In the UT region, the PMS values for all cases were located in a range of the corresponding region (215–383 hPa) by combining more than two wavelength ranges. It was also observed in the PMS values in the MT region. But in the LMT region, only the PMS values of combination of the three wavelength ranges were located in the vertical region of LMT. The PMS value for all profiles averaged in the LMT region was 783 hPa and 808 hPa for the UV+TIR and UV+TIR+MW measurements, respectively. The PMS value was increased by approximately 3 % by adding the MW measurement to the UV+TIR measurements, although the PMS value of the MW measurement itself was lower than 300 hPa in the LMT region.

The reduction rate of error, RRE, in the ozone partial column of ozone calculated using Eq. (14), is shown in the right column of Fig. 4. The value of RRE was approximately 0–20 % in the UT, MT and LMT regions. The RRE generally increased by combining more wavelength ranges in the ozone synergetic retrieval as DFS shown in Fig. 3. The RRE value for all profiles averaged in the LMT region was 9 % and 11 % for UV+TIR and UV+TIR+MW measurements, respectively. Adding the MW measurement made 2 % increase of RRE value. A certain increase of the retrieval sensitivity of the LMT ozone was shown by DFS as well as PMS and RRE.

The sensitivity of MW measurement in the UT region largely depended on the atmospheric profile used in this simulation, and its dependency transferred to the wavelength combinations including the MW measurement. The DFS values of the MW measurement in the UT region for profiles in December 2009 (green and blue markers in Fig. 3) were larger than those in June 2009 (red and purple markers). The sensitivity of the MW measurements in the UT region increased for profiles with large amounts of ozone in the UT region. In the LMT, the DFS values of the UV and TIR measurements strongly depended on the atmospheric profiles. The average value of the partial column of ozone in the LMT region at CEC in June 2009 was

$5.03 \times 10^{21}$ m$^{-2}$, the largest among the four cases. Only the DFS value of UV+TIR+MW at CEC in June 2009 (red marker) was larger than unity (1.03±0.01).

More details of the vertical characteristics are discussed with $A$. Figures 5 and 6 show $A$ and $\mathbf{m}$ obtained from the simulation using the atmospheric profile #01 and #12 for all of the wavelength combinations. The DFS values in the LMT for the UV+TIR+MW measurements of profiles #01 and #12 were respectively estimated to be 1.04 and 0.27, i.e., which were the highest and lowest values among the 20 profiles. In the case of profile #01, the UV, TIR and MW measurements provided information in the LMT, MT to UT, and UT to the stratosphere, respectively. The UV and TIR measurements were important to retrieve the ozone amount in the LMT when only one wavelength range was used because their peaks in the row of $A$ were located in the LMT region. The FWHM of the row of $A$ of the UV and TIR measurements in the LMT was approximately 3 km. The peak value in the row of $A$ increased from 0.25 of the UV and TIR measurements to 0.35 as a result of adding the two measurements. The combination of TIR and UV improved the sensitivity of retrieval of ozone in the LMT, as shown in the previous studies (Landgraf and Hasekamp, 2007; Worden et al., 2007; Natraj et al., 2011; Cuesta et al., 2013; Fu et al., 2013). Adding the MW measurement further increased it to 0.4. In profile #12 (less ozone in the LMT), the peak of the row of $A$ for the UV measurement was located in the MT (maximum value of 0.1). Although the peak value in the MT increased to 0.23 as a result of adding the TIR and MW measurements, the peak value in the LMT remained low.

Additionally, we performed sensitivity study for the DFS, PMS and RRE using several $\sigma_a$ values in order to obtain error volume. Figure 7 shows the DFS, PMS and RRE values with $\sigma_a$ of 100 %, 50 %, 30 %, 20 % and 10 %. Here the calculation results for all of 20 profiles were averaged. The DFS values decreased as the $\sigma_a$ value decreased for all wavelength combinations considered in this study. This behavior was also observed for the RRE results. The PMS value seemed to locate in the corresponding vertical region when the $\sigma_a$ value increased. The behavior of the relative differences of DFS, PMS and RRE for the different wavelength combination was the same for all $\sigma_a$ cases. It was shown that the increase of retrieval sensitivity of the LMT ozone by adding the MW measurement was observed for all $\sigma_a$ cases considered in this sensitivity study.

In this study, we showed that an introduction of the MW limb measurement had a certain effect to increase the sensitivity of the tropospheric ozone retrieval. However, following issues might cause bias and uncertainties in the retrieval results and should be considered to implement this retrieval method to real measurements. Discrepancy in spectroscopic parameters for several wavelength ranges is one of the most important error sources. For the ozone retrieval using the MW limb measurement, spectroscopic parameters are the largest error sources. It was reported that approximately 3–5 % error was caused by uncertainties in air-broadening coefficient and line intensity in the case of the SMILES observation (Kasai et al., 2013). It is comparable to the approximately 4 % uncertainty in the spectroscopic parameters in the UV and TIR wavelength ranges (Gratien et al., 2010). The tangent height correction can also be a large error source for the MW limb measurement (Kasai et al., 2013). The tangent height is a key parameter to determine the field-of-view, thus uncertainty in the tangent height causes discrepancy of the atmospheric layer assumed in the simulation and the true atmospheric layer. This discrepancy critically affects to the retrieval of the ozone amount in both the stratosphere and the troposphere, and also might cause bias for correction of time delay between the MW limb measurement and the other nadir measurements. In this study, we assumed instruments onboard the ISS (low orbit) and the time difference of approximately five minutes could be ignored. If the time difference was long and

its correction was required, three-dimensional atmospheric modeling should be performed including the field-of-view of the MW limb measurement.

As a whole, it was shown that retrieval of the tropospheric ozone was improved by adding the MW limb measurements to the UV and TIR nadir measurements. In the LMT region, the DFS value was estimated to increase about 30 %. The DFS value was

estimated to be 0.75 and 0.66 over land and ocean, respectively, for the future mission of IASI-NG and UVNS (Constantino et al., 2017). If the MW limb measurement is implemented to this synergetic retrieval, the DFS value is estimated to increase to 0.98 and 0.86 over land and ocean, respectively. Our feasibility study showed a possibility to retrieve the ozone in the LMT with a DFS value of unity.

## 5    Conclusions

We performed a feasibility study of obtaining a vertically resolved ozone profile in the troposphere from synergetic retrieval using a combination of three separate wavelength ranges (UV, TIR, and MW). Observation geometries used in this study were the nadirs for the UV and TIR measurements and limb for the MW measurement from low orbit at a height of 300 km (the height of the ISS). The urban (CEC) and ocean (ECS) area in June and December 2009 were assumed in this study. We evaluated the sensitivities of retrieval of ozone in the three vertical regions (UT (215-383 hPa), MT (383-749 hPa) and LMT

(>749 hPa)) in terms of the degree of freedom for signal (DFS) based on the OEM calculation. The pressure of maximum sensitivity (PMS), the reduction rate of error (RRE) for the partial column were also used as an indicator of the sensitivity evaluation.

The TIR measurement was most sensitive for retrieving ozone in the UT when only one wavelength range was used. The additional MW measurement was most effective at improving the sensitivity in the UT when combining several wavelength

ranges. The DFS values in the UT for all of 20 profiles averaged were 0.62±0.08 and 1.21±0.28 for the UV+TIR and UV+TIR+MW measurements, respectively. In the MT region, the contribution of the TIR measurement was dominant in the DFS calculation. The average DFS value of the TIR measurement for all profiles averaged was 0.83±0.11. It was increased to more than unity by adding either the UV or MW measurements. The UV and TIR measurements were dominant in the retrieval of ozone in the LMT region. The DFS value in the LMT strongly depended on the ozone abundance. The DFS value

in the LMT became larger in the case of larger partial column of the LMT ozone. The largest DFS value in the LMT for the UV+TIR+MW measurement of 1.03±0.01 was given by the case at CEC in June 2009 which is the case of the largest LMT ozone enhancement with the partial column in the LMT ozone of approximately $5 \times 10^{21}$ m$^{-2}$.

The MW limb measurement alone derived less information in the MT and LMT regions. The DFS values were less than 0.01 and the PMS was located at vertical region higher than the MT and LMT regions. Nevertheless, adding MW measurements to

the UV and TIR measurement combinations improved sensitivity not only in the UT but also in the MT and LMT. The DFS values were increased by 96 %, 23 % and 30 % in the UT, MT and LMT, respectively, by adding the MW measurements to the UV+TIR measurements. This might indicate that reducing the uncertainty of ozone abundance in the stratosphere is important for estimating an accurate tropospheric ozone profile.

*Acknowledgements.* This work was supported by the Funding Program for Next Generation World-Leading Researchers (NEXT Program) (No. GR101). This research was partly supported by Coordination Funds for Promoting Space Utilization by the Ministry of Education, Culture, Sports, Science and Technology (MEXT), Japan. We are grateful to Dr. K. Takigawa for providing us with his numerical simulation data for this study. The MERRA data used in this study was provided by the Global Modeling and Assimilation Office (GMAO) at NASA Goddard Space Flight Center through the NASA GES DISC online archive. We made use of CIRA-86. We thank Dr. K. Kikuchi for discussing with us the specifications of the MW instrument proposed for the Air Pollution Observations from the ISS. We thank our colleagues in the Aura MLS team (JPL) for valuable discussion on the frequency selection of the APOLLO MW instrument. We are grateful to Dr. A. Rozanov for giving a lot of useful comments of SCIATRAN.

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

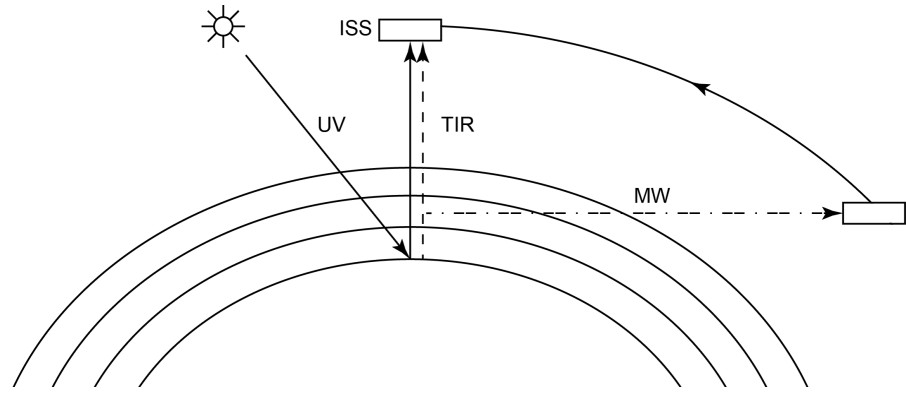

**Figure 1.** Geometries of down-looking nadir (UV and TIR) and limb (MW) observations.

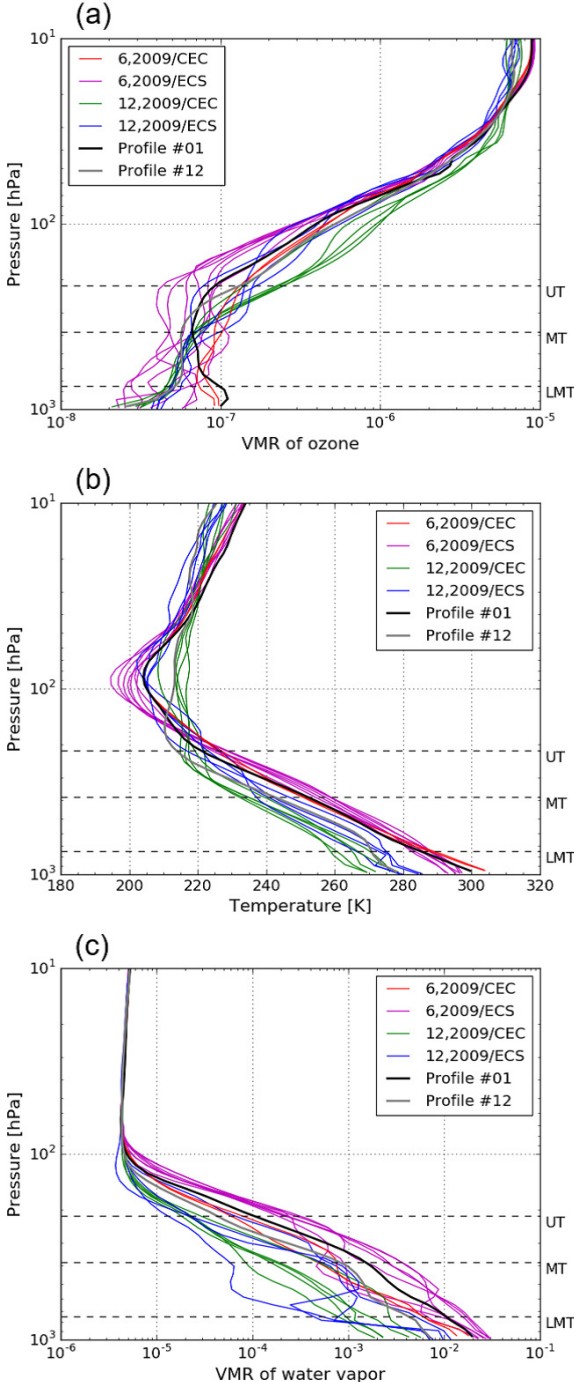

**Figure 2.** The 20 atmospheric scenarios used in this study: (a) VMR of ozone, (b) temperature, and (c) VMR of water vapor. These atmospheric scenarios can be divided into four groups, as denoted by four different color curves: (red) June 2009 in CEC, (purple) June 2009 in ECS, (green) December 2009 in CEC, and (blue) December 2009 in ECS. Two example profiles #01 and #12, represented by the black and gray lines, respectively, are the ones used to obtain the results shown in Figs. 5 and 6.

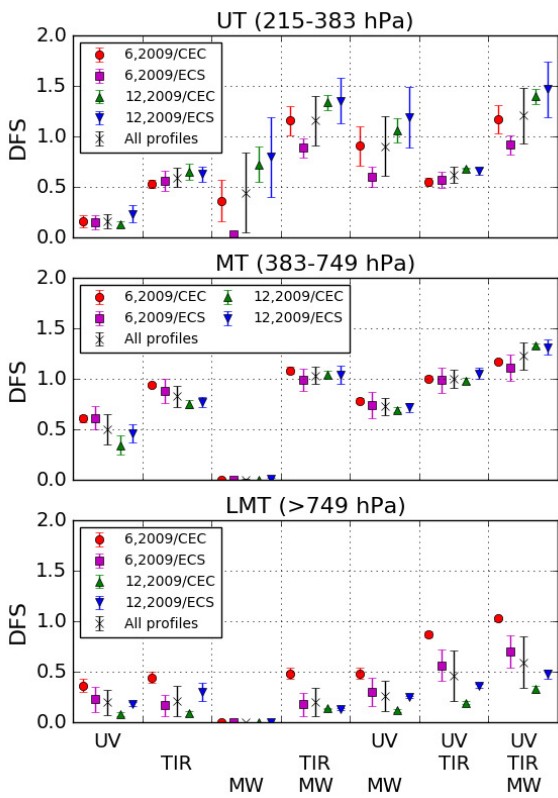

**Figure 3.** Values of DFS in the upper troposphere (UT, 215–383 hPa), middle troposphere (MT, 383–749 hPa), and lowermost troposphere (LMT, >749 hPa): (red) June 2009 in CEC, (purple) June 2009 in ECS, (green) December 2009 in CEC, (blue) December 2009 in ECS and (black) all of 20 profiles.

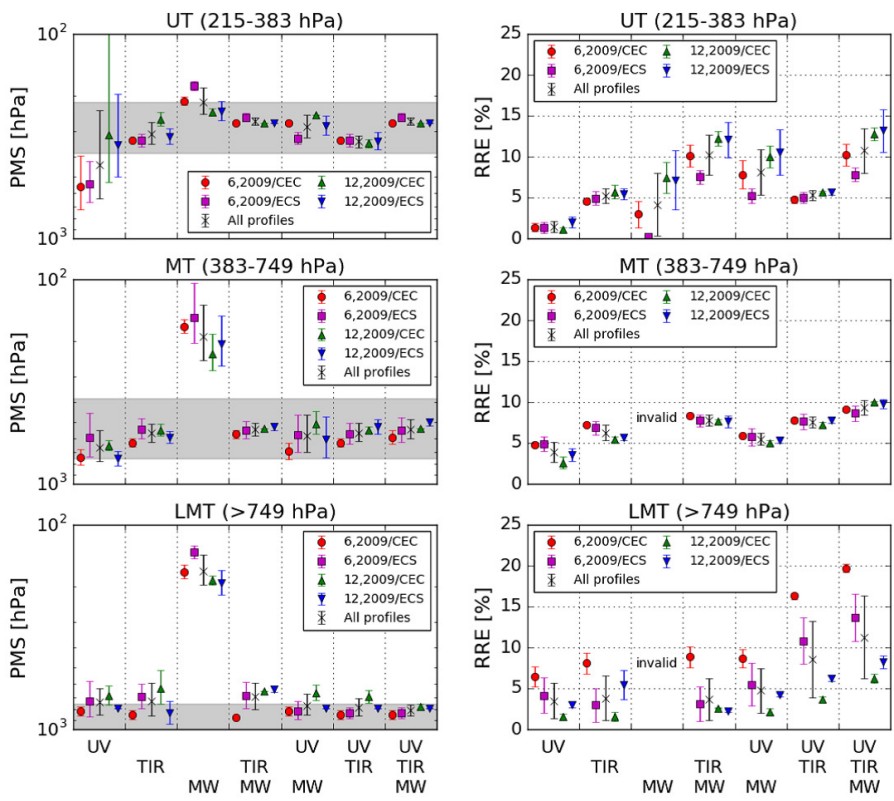

**Figure 4.** Same as Fig. 3 but for the PMS (left) and RRE (right). The gray shaded area represents the vertical region that corresponds to UT, MT and LMT.

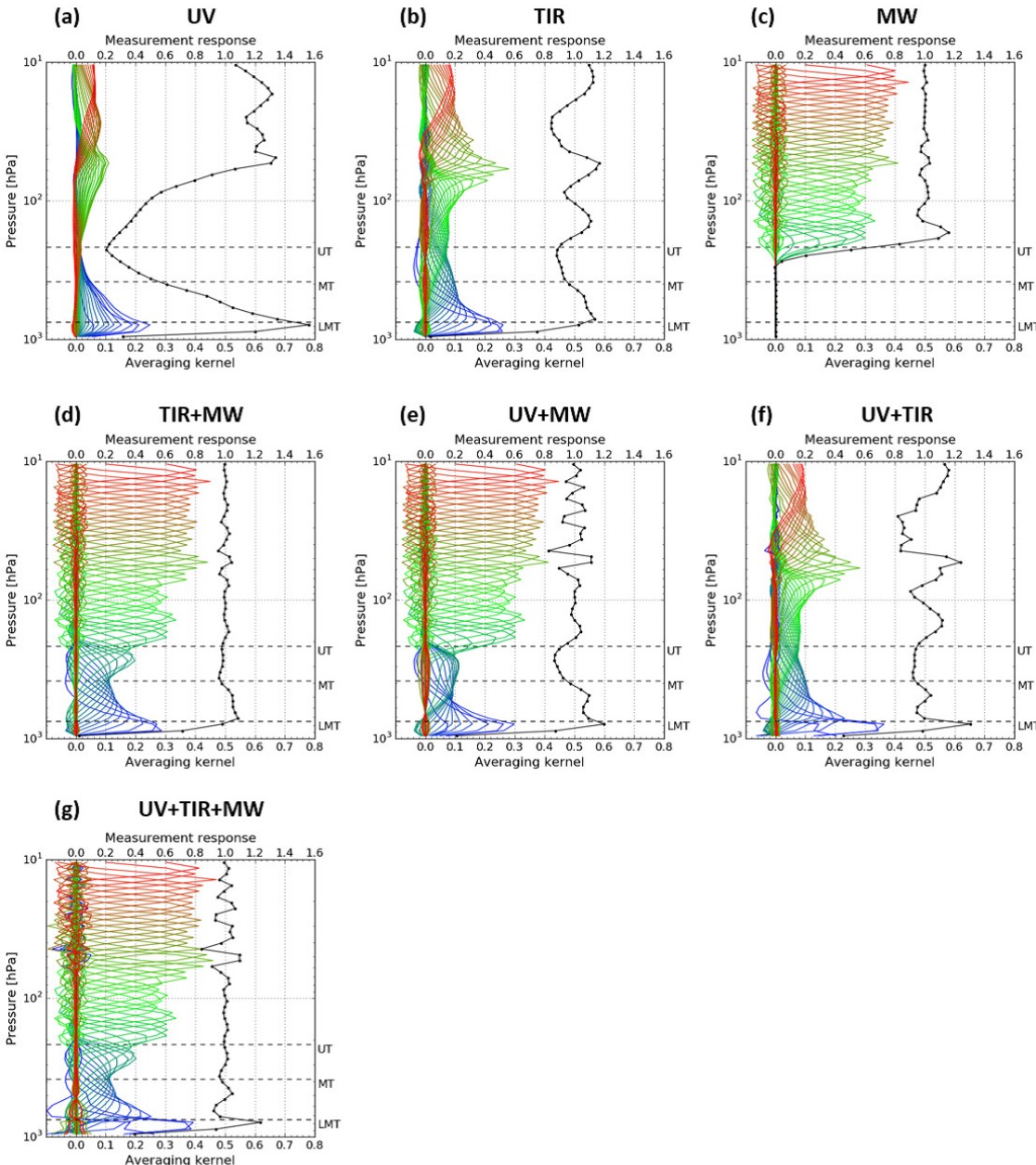

**Figure 5.** Averaging kernels of the atmospheric profile #01 (June 2009 in CEC) for seven wavelength selection cases: (a) UV, (b) TIR, (c) MW, (d) UV+TIR, (e) UV+MW, (f) TIR+MW, and (g) UV+TIR+MW. The measurement responses are also plotted by black dots and lines. Atmospheric profile #01 is a scenario which has an ozone enhancement in the LMT.

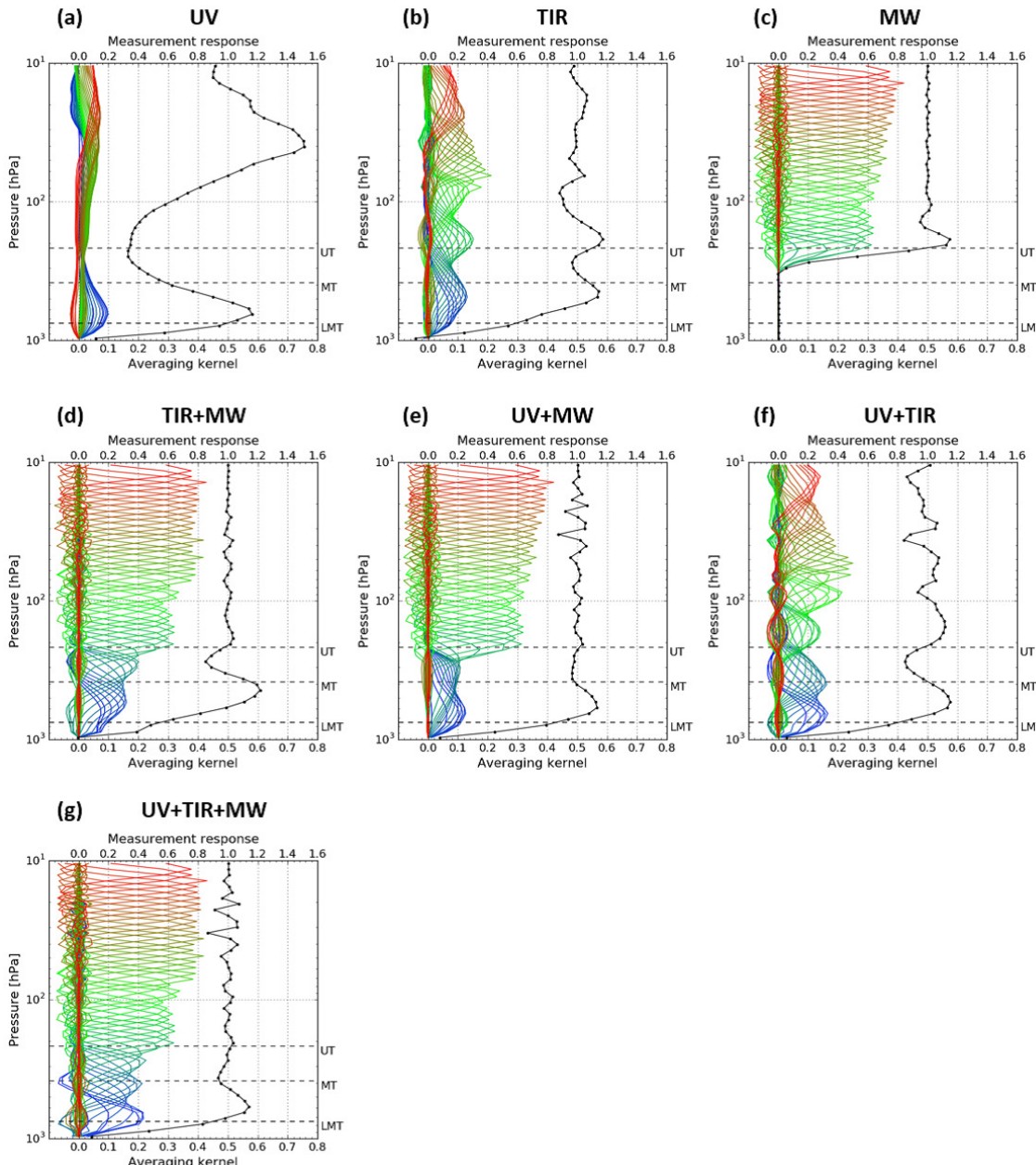

**Figure 6.** Same as Fig. 5 but for atmospheric profile #12 (December 2009 in CEC). This profile is a scenario that there is less ozone in the LMT.

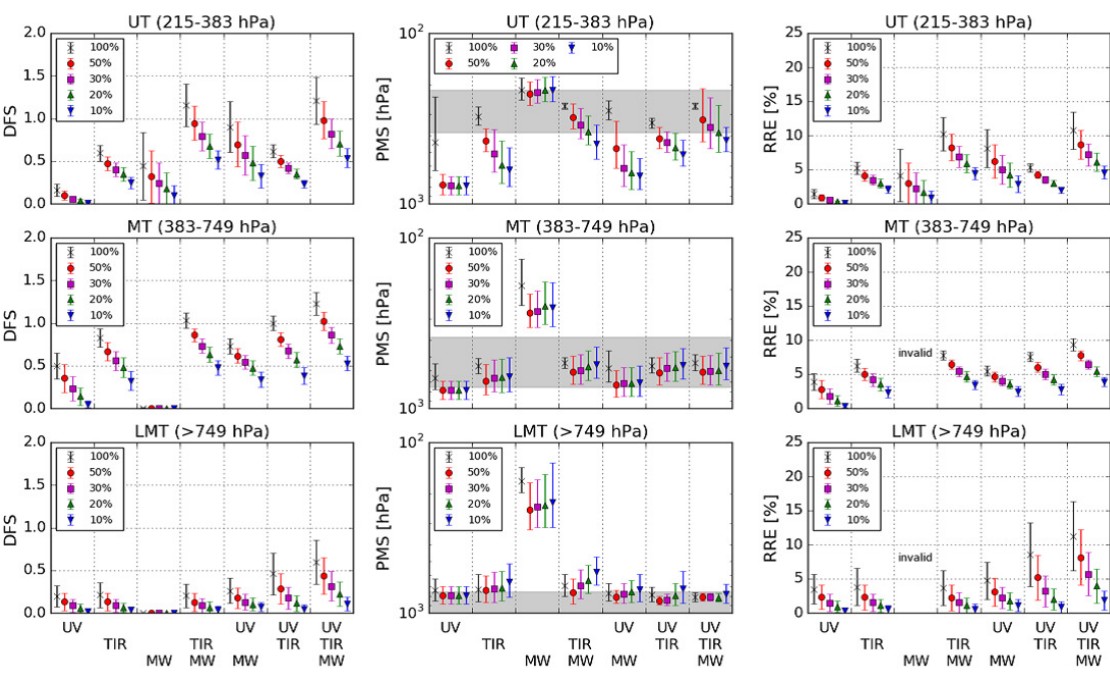

**Figure 7.** Values of DFS, PMS and RRE in the upper troposphere (UT, 215–383 hPa), middle troposphere (MT, 383–749 hPa), and lowermost troposphere (LMT, >749 hPa) for all of 20 profiles averaged with different $\sigma_a$ settings. The value of $\sigma_a$ $ranges$ $to$ 100 % (black), 50 % (red), 30 % (purple), 20 % (green) and 10 % (blue) of the log-based a priori VMR.

**Table 1.** Specifications of the three assumed instruments and the radiative transfer models used in this study.

| | UV | TIR | MW |
|---|---|---|---|
| Observation geometry | Nadir-viewing | Nadir-viewing | Limb-viewing |
| Wavelength | 305–340 nm | 980–1080 cm$^{-1}$ | 345–357 GHz, 639–651 GHz |
| Spectral resolution | 0.6 nm | ~0.12 cm$^{-1}$ | 25 MHz |
| Sampling step | 0.2 nm | ~0.12 cm$^{-1}$ | 25 MHz |
| Sensitivity[a] | 90 (305 nm)–1400 (340 nm) | 300 | 0.7 K (350 GHz band), 1.7 K (645 GHz band) |
| Scattering | Yes | No | No |
| Emission | No | Yes | Yes |
| Forward model | SCIATRAN | LBLRTM | AMATERASU |

[a] Instrumental sensitivity is described in the commonly used way for each spectral region; that is, SNR for UV and TIR and noise equivalent brightness temperature for MW.

**Table 2.** Estimation of three reference SNR value in the UV simulation.

|  | SNR at 305 nm | SNR at 340 nm |
|---|---|---|
| Case 1: high level radiance (albedo 90 %, SZA 0°) | ~200 | ~2550 |
| Case 2: middle level radiance (albedo 25 %, SZA 45°) | ~60 | ~1200 |
| Case 3: low level radiance (albedo 5 %, SZA 80°) | ~10 | ~450 |

**Table 3.** Summary of 20 atmospheric scenarios used in the simulation.

| # | Date[a] | Area[b] | $T_s^c$ [K] | $P_s^c$ [hPa] | PC (UT)[d] [m$^{-2}$] | PC (MT)[d] [m$^{-2}$] | PC (LMT)[d] [m$^{-2}$] | $H_2O^e$ [g/cm$^2$] |
|---|---|---|---|---|---|---|---|---|
| 01 | 6/16 | CEC | 301.4 | 976.0 | $2.57\times10^{21}$ | $5.87\times10^{21}$ | $5.66\times10^{21}$ | 3.4 |
| 02 | 6/24 | CEC | 304.5 | 970.2 | $4.13\times10^{21}$ | $6.14\times10^{21}$ | $4.51\times10^{21}$ | 2.1 |
| 03 | 6/25 | CEC | 305.0 | 970.2 | $3.65\times10^{21}$ | $6.52\times10^{21}$ | $4.93\times10^{21}$ | 2.3 |
| 04 | 6/3 | ECS | 293.9 | 999.6 | $1.49\times10^{21}$ | $4.39\times10^{21}$ | $3.72\times10^{21}$ | 4.5 |
| 05 | 6/9 | ECS | 294.9 | 1010.9 | $1.68\times10^{21}$ | $4.17\times10^{21}$ | $3.96\times10^{21}$ | 4.2 |
| 06 | 6/20 | ECS | 296.0 | 1004.2 | $2.92\times10^{21}$ | $6.34\times10^{21}$ | $1.86\times10^{21}$ | 3.7 |
| 07 | 6/21 | ECS | 296.5 | 1002.9 | $3.00\times10^{21}$ | $4.26\times10^{21}$ | $2.39\times10^{21}$ | 6.0 |
| 08 | 6/26 | ECS | 296.8 | 1010.0 | $3.27\times10^{21}$ | $6.20\times10^{21}$ | $2.25\times10^{21}$ | 4.1 |
| 09 | 6/27 | ECS | 297.6 | 1006.6 | $2.13\times10^{21}$ | $3.13\times10^{21}$ | $1.49\times10^{21}$ | 6.0 |
| 10 | 6/30 | ECS | 298.1 | 1004.5 | $2.71\times10^{21}$ | $3.05\times10^{21}$ | $1.80\times10^{21}$ | 5.9 |
| 11 | 12/2 | CEC | 280.7 | 993.1 | $3.20\times10^{21}$ | $4.47\times10^{21}$ | $1.83\times10^{21}$ | 1.1 |
| 12 | 12/11 | CEC | 280.0 | 988.8 | $2.66\times10^{21}$ | $4.38\times10^{21}$ | $2.10\times10^{21}$ | 1.4 |
| 13 | 12/20 | CEC | 271.6 | 997.3 | $4.58\times10^{21}$ | $4.14\times10^{21}$ | $2.22\times10^{21}$ | 0.3 |
| 14 | 12/22 | CEC | 278.1 | 985.9 | $4.25\times10^{21}$ | $4.27\times10^{21}$ | $2.21\times10^{21}$ | 0.6 |
| 15 | 12/27 | CEC | 271.3 | 992.6 | $4.51\times10^{21}$ | $4.41\times10^{21}$ | $2.10\times10^{21}$ | 0.4 |
| 16 | 12/28 | CEC | 274.0 | 985.8 | $4.12\times10^{21}$ | $4.33\times10^{21}$ | $2.15\times10^{21}$ | 0.4 |
| 17 | 12/4 | ECS | 286.7 | 1019.2 | $4.04\times10^{21}$ | $4.48\times10^{21}$ | $2.54\times10^{21}$ | 1.1 |
| 18 | 12/12 | ECS | 287.9 | 1019.9 | $2.35\times10^{21}$ | $4.81\times10^{21}$ | $2.92\times10^{21}$ | 2.0 |
| 19 | 12/21 | ECS | 282.4 | 1025.7 | $4.91\times10^{21}$ | $4.74\times10^{21}$ | $2.68\times10^{21}$ | 0.8 |
| 20 | 12/26 | ECS | 281.3 | 1019.3 | $3.72\times10^{21}$ | $4.38\times10^{21}$ | $2.68\times10^{21}$ | 0.9 |

[a] All simulation data are from 2009.

[b] CEC and ECS stand for Central East China (30°N–40°N, 110°E–123°E) and the East China Sea (29°N–33°N, 125°E–129.5°E), respectively.

[c] Temperature and pressure at the surface.

[d] PC means ozone partial column. PC is presented for each altitude region: upper troposphere (UT, 215–383 hPa), middle troposphere (MT, 383–749 hPa), and lowermost troposphere (LMT, > 749 hPa).

[e] $H_2O$ column amount in the troposphere.

**Table 4.** Mean and standard deviation (1-$\sigma$) of DFS in the upper troposphere (UT, 215–383 hPa), middle troposphere (MT, 383–749 hPa), and lowermost troposphere (LMT, >749 hPa). The standard deviation is listed in the parenthesis. For example, 0.16(6) means 0.16 of the mean value and 0.06 of the standard deviation.

|  |  | UV | TIR | MW | TIR+MW | UV+MW | UV+TIR | UV+TIR+MW |
|---|---|---|---|---|---|---|---|---|
| 6, 2009/CEC | UT | 0.16(6) | 0.53(4) | 0.36(20) | 1.16(14) | 0.91(20) | 0.55(4) | 1.17(14) |
|  | MT | 0.61(3) | 0.94(2) | <0.01 | 1.08(4) | 0.78(3) | 1.00(2) | 1.17(1) |
|  | LMT | 0.37(6) | 0.44(6) | <0.01 | 0.48(6) | 0.48(5) | 0.87(3) | 1.03(1) |
| 6, 2009/ECS | UT | 0.15(7) | 0.56(10) | 0.03(3) | 0.89(9) | 0.60(10) | 0.57(8) | 0.92(10) |
|  | MT | 0.61(11) | 0.88(12) | <0.01 | 0.99(11) | 0.74(13) | 0.99(13) | 1.11(13) |
|  | LMT | 0.23(13) | 0.17(11) | <0.01 | 0.18(11) | 0.30(14) | 0.56(15) | 0.70(16) |
| 12, 2009/CEC | UT | 0.12(3) | 0.65(8) | 0.72(17) | 1.34(8) | 1.06(12) | 0.68(1) | 1.40(7) |
|  | MT | 0.34(9) | 0.75(4) | <0.01 | 1.04(3) | 0.69(2) | 0.98(3) | 1.33(2) |
|  | LMT | 0.08(2) | 0.09(3) | <0.01 | 0.14(1) | 0.12(2) | 0.19(1) | 0.33(3) |
| 12, 2009/ECS | UT | 0.23(8) | 0.63(8) | 0.80(40) | 1.35(23) | 1.19(30) | 0.66(4) | 1.47(27) |
|  | MT | 0.46(9) | 0.77(5) | <0.01 | 1.04(9) | 0.72(5) | 1.05(5) | 1.31(7) |
|  | LMT | 0.18(2) | 0.30(9) | <0.01 | 0.13(1) | 0.25(1) | 0.36(2) | 0.48(5) |
| All profiles | UT | 0.16(7) | 0.59(9) | 0.44(40) | 1.16(25) | 0.90(30) | 0.62(8) | 1.21(28) |
|  | MT | 0.50(15) | 0.83(11) | <0.01 | 1.03(8) | 0.73(9) | 1.00(9) | 1.23(13) |
|  | LMT | 0.20(12) | 0.21(15) | <0.01 | 0.20(14) | 0.26(15) | 0.46(25) | 0.59(26) |

**Table 5.** Comparison of DFS in the LMT region with previous studies. The method and situation to derive the DFS value are different for each study, thus, the relative difference between the DFS value for the UV+TIR measurement ($DFS_{UV+TIR}$) and mean of DFS values for UV and TIR measurements ($DFS_{UV}$, $DFS_{TIR}$) was used for this comparison. The relative difference was calculated by ($DFS_{UV+TIR}$ − Mean($DFS_{UV}$, $DFS_{TIR}$))/Mean($DFS_{UV}$, $DFS_{TIR}$).

|  | $DFS_{UV}$ | $DFS_{TIR}$ | Mean($DFS_{UV}$, $DFS_{TIR}$) | $DFS_{UV+TIR}$ | Rel. dif. | Definition of LMT |
|---|---|---|---|---|---|---|
| This work | 0.20 | 0.21 | 0.203 | 0.458 | 126 % | >749 hPa |
| Fu et al. (2013) | 0.10 | 0.21 | 0.15 | 0.37 | 139 % | >700 hPa |
| Cuesta et al. (2013)[a] | 0.08 | 0.20 | 0.14 | 0.285 | 104 % | <3 km |
| Natraj et al. (2011) | 0.26 | 0.27 | 0.265 | 0.57 | 115 % | >800 hPa |

[a] DFS values over land and ocean are averaged.