# Peer review of "Vertical profile of tropospheric ozone derived from synergetic retrieval using three different wavelength ranges, UV, IR, and Microwave: sensitivity study for satellite observation"

_Atmospheric Measurement Techniques, 2017_

## Short Comment (SC1) · 31 May 2017

Dear Kasai et al.,

I have found your manuscript of a huge interest, as it further extends the path towards multi-spectral integration in retrieval algorithms and it enables significantly improved vertical information extraction on tropospheric ozone.

Nevertheless, in your introduction I have found that you have missed what has been the first experimental step (i.e. using real satellite data) of all this spectral integration. Consequently, I suggest adding the following references of our past work:

[Figure]

Sellitto, P., Del Frate, F., Solimini, D., and Casadio, S.: Tropospheric ozone column retrieval from ESA-Envisat SCIAMACHY nadir UV/VIS radiance measurements by means of a neural network algorithm, IEEE T. Geosci. Remote, 50, 998–1011, doi:10.1109/TGRS.2011.2163198, 2012

&

Sellitto, P., Di Noia, A., Del Frate, F., Burini, A., Casadio, S., and Solimini, D.: On the role of visible radiation in ozone profile retrieval from nadir UV/VIS satellite measurements: An experiment with neural network algorithms inverting SCIAMACHY data, J. Quant. Spectrosc. Ra., 113, 1429–1436, doi:10.1016/j.jqsrt.2012.04.007, 2012

that first introduced the combined use of UV plus visible (i.e. the ozone Chappuis absorption bands) spectral information and demonstrated that it improves vertical sensitivity of these retrievals. This was the first use of different bands, even if in a single satellite instrument, and is of a historical importance in this path that you are spectacularly extending using MW information, after the works of Cuesta et al and Fu et al that introduced the use of TIR spectral information.

Thank you for your very interesting work. Pasquale Sellitto

---

## Short Comment (SC2) · 21 Aug 2017

Dear Dr. Sellitto

We would like to thank Dr. Sellitto for pointing out missing of references in our manuscript. Sellitto et al., 2012a, b performed tropospheric ozone retrieval using the SCIAMACHY nadir UV and VIS measurements. They used a new method based on neural network technique, and showed a significant improvement of the tropospheric ozone retrieval by using the UV and VIS measurements. These works should be cited

in our paper as you mentioned. We modified our manuscript as below.

Page 2 Line 33 "the tropospheric ozone retrieval using the optimal estimation method (OEM) (Rodgers, 2000)" was added.

Page 3 Lines 7 – 10 "The other approach to retrieve the tropospheric ozone profile using neural network technique was performed with the SCIAMACHY nadir measurements in the UV and VIS ranges (Sellitto et al., 2012a, b). They also showed a significant availability of combining several wavelength ranges to retrieve the tropospheric ozone profile." was added.

Page 4 Lines 10 – 13 "The VIS (340-505 nm) and shorter UV spectral region (< 305 nm) were not taken into account, because little information on surface ozone can be extracted from measurements in these ranges (e.g., Bak et al., 2012). Moreover, the wavelength dependence of the surface reflectance, absorption of NO2 and the Ring effect were out of the scope of the study." → "We decided not to include the VIS (340-505 nm) and the shorter UV spectral range (< 305 nm) in this study, although the benefit of adding VIS wavelengths has been reported (Sellitto et al., 2012a, b). The reason why we excluded these ranges is because the wavelength dependence of the surface reflectance, absorption of NO2 and the Ring effect were out of the scope of the study."

References "Sellitto, P., Del Frate, F., Solimini, D., and Casadio, S.: Tropospheric ozone column retrieval from ESA-Envisat SCIAMACHY nadir UV/VIS radiance measurements by means of a neural network algorithm, IEEE Transactions on Geoscience and Remote Sensing, 50, 998–1011, doi:10.1109/TGRS.2011.2163198, 2012a" was added.

"Sellitto, P., Di Noia, A., Del Frate, F., Burini, A., Casadio, S., and Solimini, D.: On the role of visible radiation in ozone profile retrieval from nadir UV/VIS satellite measurements: An experiment with neural network algorithms inverting SCIAMACHY data, Journal of Quantitative Spectroscopy and Radiative Transfer, 113, 1429–1436, doi:10.1016/j.jqsrt.2012.04.007, 2012b" was added.
"Bak, J., Kim, J. H., Spurr, R. J. D., Liu, X., and Newchurch, M. J.: Sensitivity study of ozone retrieval from UV measurements on geostationary platforms, Remote Sensing of Environment, 118, 309–319, 2012." was removed.

Sincerely yours,

Tomohiro Sato and Yasuko Kasai

National Institute of Information and Communications Technology

Please also note the supplement to this comment:
https://www.atmos-meas-tech-discuss.net/amt-2017-98/amt-2017-98-SC2-supplement.pdf

**Supplement:**

[revised manuscript text omitted]

---

## Referee Comment (RC1) · Anonymous Referee #1 · 1 Sep 2017

This paper presents the results of a study designed to facilitate the combination of space borne spectral measurements in the ultraviolet, thermal infrared and microwave wavelength ranges to determine ozone concentration in the tropospheric/lower stratosphere. The study combines nadir viewing (UV and TIR) with limb-sounding (MV) remote sounding techniques. Standard optimal estimation retrieval theory is used to evaluate the expected performance of this triad of remote sounding measurements. Results presented include degrees of freedom for signal, partial column errors and elements of the averaging kernel matrices for a range of East Asian summer and winter

atmospheric conditions. Although the MW instrument itself has no sensitivity below the middle troposphere, the MW provides additional constraints in the stratosphere that actually benefit the retrievals of ozone in the lowermost troposphere than compared to using UV+TIR alone.

The paper is well written and the material is presented in a clear manner. My main concern though is the extent to which the study is applicable to real world instruments, since for example the complications arising from clouds are not considered and neither are the inevitable inconsistencies of instrument biases, both which of course are the bane of researchers trying to perform multi-instrument simultaneous retrievals. I would have liked at least some discussion on these points. Aerosols are mentioned in connection with the UV instrument, but it is not clear if these are in a layer below the sensitivity of the TIR instrument.

I recommend publication of this paper in AMT.

Minor corrections.

P4,L20: MLS also has other ozone bands (in the 190GHz, 640GHZ and THz radiometers) in addition to the standard product at 240GHz.

P4,L30: This seems to be a suggesting 17.5 second vertical limb scan. It would be worthwhile to point this out in connection with Fig 1. Also the paper state that the time delay between nadir and limb views (5 mins) has been ignored. Is that because the time difference has negligible effect on the atmospheric scenes?

P7,L28: There is almost certainly covariance in the apriori, but you are using none. Have you done any simulations with off-diagonal components in Sa to see how the results are affected?

P9: A lot of values are given in the text. Could these be placed in a table?

---

## Referee Comment (RC2) · Anonymous Referee #2 · 5 Sep 2017

The paper by Kasai et al. presents an interesting simulation study of the potential of a synergetic retrieval using UV, IR and microwave (UV+IR+MW) spectra for retrieving tropospheric ozone vertical profiles. This multispectral approach could be implemented in future Japanese air-quality monitoring missions. Whereas the topic and relevance of the paper are suitable for a possible publication at AMT, the current manuscript needs substantial revision in order to add and complete with important information and correct some inconsistencies of the results. As described below, part of the calculations are needed to be done again (UV radiative transfer and error/averaging kernels

estimations) and many aspects should be more detailed (comparisons with previous multispectral methods, analysis of sensitivity as function of height, error estimations, etc.).

The major revisions needed for the manuscript are the following:

1) The choice of some of the parameters used in the simulation are neither clearly justified nor realistic: in page 3 lines 41-43, it is mentioned that a surface albedo of 90% and solar zenith angle of 0° are considered. These parameters are of course not realistic. Indeed, surface albedo at the UV is often from 5% to 15% and only greater than 30% for exceptionally bright surfaces. Solar zenith angles are also key parameters affecting UV spectra that greatly vary depending on overpass time and location of the measurement. Simulations of UV spectra and jacobians should be done again with realistic values of surface albedo and diverse solar zenith angles. The use of appropriate values is key for a proper estimation of the performance of the retrieval approach and they should be realistic for each scene observed.

2) Retrieval error estimations show extremely high levels (i.e. PCE from Figure 3). These values (around 65%) are too high for a proper retrieval. This is the case for all retrievals using diverse spectral domains (even for single band retrievals). We expect typical retrieval errors for tropospheric ozone around 20 to 30% (IR, UV, etc) and NOT 65%. Retrieval errors of diverse tropospheric ozone columns for most single band retrievals and even UV+IR approaches implemented to real data are indeed of 20-30%. This aspect should be revised in the simulations presented by Kasai et al.. For example, retrievals can be more constrained for obtaining lower errors and therefore will provide lower sensitivity. Sensitivity is then overestimated when accepting too high errors. Therefore, the retrieval approach used in the paper to estimate averaging kernels and errors in the results presented in the paper (Figures 3, 4 and 5) should be performed again.

3) Explicit comparison with other multispectral synergisms: The added value of the
method proposed in the paper should gain great clarity with an explicit comparison with previous work on multispectral retrieval of ozone. The performance of the UV+IR+MW synergism (sensitivity and probed height) and characteristics of the instruments (spectral resolution, signal-to-noise ratio) considered in the paper should be compared to previous multispectral approaches to retrieve ozone from space. This comparison should be done with the multispectral methods implemented with real measurements (OMI and TES, Fu et al., 2013 and GOME-2 and IASI, Cuesta et al. 2013) and also future approaches (e.g. UVNS and IASI-NG, Costantino et al., 2017).

4) Heights of maximum sensitivity for each partial column: in the paper, only the degrees of freedom for the retrieval of ozone partial column are used as diagnostic for sensitivity. However, averaging kernels explicitly provide information on the height of maximum sensitivity that varies for each retrieval method and measuring conditions. Therefore, I strongly recommend to provide the heights of maximum sensitivity for each ozone partial column retrieved from satellite spectra, and compare them explicitly for each spectral combination (UV, IR, MW, UV+IR+MW, etc)..

5) Additional terms for error estimations: for such a new multispectral retrieval, I strongly recommend to estimate the contribution of additional terms of errors such as cross errors from joint fit of surface albedo and emissivity and other systematic errors (atmospheric and surface temperature, water vapor, etc).

6) A better description of the atmospheric scenario is needed. How model and airborne data are combined? How the 20 cases are chosen? What is the main characteristic (time of the day, region, ozone load, surface and atmospheric properties) of each scenario? These aspects should be explicitly commented and described in the text of the paper.

7) Spectroscopy coherence should be mentioned as an important issue for real retrievals. There exist previous multispectral retrievals combining microwave spectra with other domains? Spectroscopic consistency has been analyzed as for UV and IR (e.g.

Gratien et al., 2010) ?

8) Aerosol in the UV: how are they accounted in simulations? Are aerosols jointly fitted or considered as perfectly known? The error of this assumption should be explicitly analysed and quantified.

9) In Fig. 3, why DFS for IR+MW are lower than for IR only at the LMT???? Please justify and verify calculations.

Additional corrections:

1) In page 2, lines 61-65: Here the performance of the combination of is done for that of OMI and TES. Similar characteristic should be provided for the synergism of IASI and GOME2 measurements, in terms of DFS, errors and altitude of maximum sensitivity.

2) Page 3: please provide the units for NESR.

3) Page 3: The UV domain <305 nm does provide useful information on stratospheric ozone. Please clarify why it is not used in the retrieval.

4) Page 3 : reference Boynard et al 2009 for IASI is not appropriate. Please use Clerbaux et al 2009 instead.

5) Page 7, lines 1-3 : Also Fu et al 2013 and Cuesta et al 2013, Landgraf et al 2007 showed the advantages of a combined UV+IR retrieval of ozone. Please add these references.

---

## Author Comment (AC2) · 23 Nov 2017

Dear Referee #2

We would like to thank the Referee #2 for the valuable comments and suggestions. Please find the manuscript with major revision. We answered to your comments and suggestions in point by point. We also improved the abstract and conclusion to make the purpose of this paper clear. We hope that current manuscript is significant for the publication in AMT. We still wonder our English is not perfect as a manuscript of AMT

even after the Native English checking. We will ask to our colleague, who is native English speaker, only once before the final version, since we do not want to disturb him frequently.

Sincerely yours,

Tomohiro Sato and Yasuko Kasai National Institute of Information and Communications Technology

Please also note the supplement to this comment:
https://www.atmos-meas-tech-discuss.net/amt-2017-98/amt-2017-98-AC2-supplement.pdf

**Supplement:**

**Reply to Referee #2**

Dear Referee #2

We would like to thank the Referee #2 for the valuable comments and suggestions. Please find the manuscript with major revision. We answered to your comments and suggestions in point by point. We also improved the abstract and conclusion to make the purpose of this paper clear. We hope that current manuscript is significant for the publication in AMT. We still wonder our English is not perfect as a manuscript of AMT even after the Native English checking. We will ask to our colleague, who is native English speaker, only once before the final version, since we do not want to disturb him frequently.

Sincerely yours,

Tomohiro Sato and Yasuko Kasai
National Institute of Information and Communications Technology

Note
**GC**: General comments
**MC**: Minor corrections or Additional corrections

**Answers to Referee #2**

**Overview comment**

The paper by Kasai et al. presents an interesting simulation study of the potential of a synergetic retrieval using UV, IR and microwave (UV+IR+MW) spectra for retrieving tropospheric ozone vertical profiles. This multispectral approach could be implemented in future Japanese air-quality monitoring missions. Whereas the topic and relevance of the paper are suitable for a possible publication at AMT, the current manuscript needs substantial revision in order to add and complete with important information and correct some inconsistencies of the results. As described below, part of the calculations are needed to be done again (UV radiative transfer and error/averaging kernels estimations) and many aspects should be more detailed (comparisons with previous multispectral methods, analysis of sensitivity as function of height, error estimations, etc.).

The major revisions needed for the manuscript are the following:

**Answer to Overview comment**

We would like to thank you for the helpful comments and suggestions. We carefully checked our calculation script according to your suggestion and found a mistake to estimate the increase ratio of DFS values. The increase ratio was modified to 30% from 40% in the LMT region. Thank you so much for this suggestion. We answered the comments point-by-point and revised according to your suggestions. We also improved the abstract and conclusion in accordance with your suggestions.

**Revisions to Overview comment**

[revised manuscript text omitted]

**GC 2-1**

The choice of some of the parameters used in the simulation are neither clearly justified nor realistic: in page 3 lines 41-43, it is mentioned that a surface albedo of 90% and solar zenith angle of 0º are considered. These parameters are of course not realistic. Indeed, surface albedo at the UV is often from 5% to 15% and only greater than 30% for exceptionally bright surfaces. Solar zenith angles are also key parameters affecting UV spectra that greatly vary depending on overpass time and location of the measurement. Simulations of UV spectra and jacobians should be done again with realistic values of surface albedo and diverse solar zenith angles. The use of appropriate values is key for a proper estimation of the performance of the retrieval approach and they should be realistic for each scene observed.

**Answer to GC 2-1**

We thank the referee for giving us this valuable comment. The statements in our manuscript were misleading, as you mentioned. In our simulation, the surface albedo was assumed to be 0.056 for CEC June, 0.063 for CEC December, 0.065 for ECS June, and 0.084 for ECS December, respectively, as described in Page 7 Lines 5 – 6. These values were obtained from the database that contains the monthly global map of the Earth's surface Lambertian equivalent reflectance (LER) deduced from the Aura/OMI measurements [Kleipool et al., 2008].

The surface albedo value of 90% that you pointed out was used only for estimating the noise equivalent spectral radiance (NESR). The NESR was calculated by dividing the simulated backscattered radiance by the signal-to-noise ratio (SNR). The SNR was estimated by a linear interpolation with the three radiances (the brightest, middle and the smallest) [Private communication with K. Gerilowski]. The smallest radiance was calculated with the condition of 90% albedo and $0^o$ SZA (cloudy condition). The middle radiance was done in 25% albedo and $45^o$ SZA. The largest radiance was done in 5% albedo and $80^o$ SZA. Figure GC2.1.1 shows the results of the SNR simulations. The SNR values were approximately 90 and 1400 (mean of the three spectra) at 305 nm and 340 nm, respectively.

The solar zenith angle (SZA), which corresponds to local time, is also a key parameter for simulation of the UV radiance, as you mentioned. The aim of this article is to investigate the sensitivity of the tropospheric ozone retrieval by adding the MW limb measurement to the synergetic retrieval approach. To purely bring out the improvement of the ozone retrieval sensitivity by combining several wavelength ranges, we performed the simulation in an ideal condition. Landgraf and Hasekamp (2007) showed that the increase of the ozone retrieval sensitivity was largest in the condition of small to moderate SZA (around noon). Therefore, we set the local time to be 11:46am and 00:29pm (around noon) for CEC and ECS, respectively. These local times correspond to the time when the surface ozone is likely increased because the photo-chemistry becomes active and the sensitivity of the ozone retrieval probably becomes high. We thought the local time condition in our simulation was sufficient for investigating the ozone retrieval sensitivity with combination of UV, TIR and MW in an ideal condition. To make it clear, we improved the statements in our manuscript as follows.

Reference
- Landgraf, J., and Hasekamp, O. P.: Retrieval of tropospheric ozone: The synergistic use of thermal infrared emission and ultraviolet reflectivity measurements from space, Journal of Geophysical Research: Atmospheres, 112, 2007.

[Figure]

Figure GC2.1.1: Estimation of SNR in the UV wavelength range. The left column figures show the radiance spectra, and the right ones show the SNR. The top and bottom panels show the results with a linear and log scales, respectively.

**Revisions to GC 2-1**

Page 4 Lines 23 – 27

"The noise equivalent spectral radiance (NESR) for a surface albedo of 90% and the solar zenith angle of 0° were approximately 90 and 2600 at 300 and 340 nm, respectively (private communication with K. Gerilowski). The NESR value was obtained by dividing the simulated backscattered radiance by the signal-to-noise ratio (SNR)."

→

"The noise equivalent spectral radiance (NESR) was obtained by dividing the simulated backscattered radiance by the signal-to-noise ratio (SNR). The SNR values were estimated by an interpolation with the small, middle and large values of the radiance calculated in the conditions that the surface albedo and solar zenith angle (SZA) of 90% and 0°, 25% and 45°, and 5% and 80°, respectively [Private communication with K. Gerilowski]. The mean values of SNR were estimated to be approximately 90 and 1400 at 305 nm and 340 nm, respectively."

Page 5 Lines 22 – 24

"The local times around noon were set because it was shown that the ozone retrieval sensitivity was most increased in small to moderate solar zenith angles (SZAs) in the simulation study performed by Landgraf and Hasekamp, (2007)." was added.

P 24 Table 1

"~90 (300 nm), ~2600 (340 nm)"

→

"90 (300 nm) – 1400 (340 nm)"

P 24 Table 1

"[b] In the case that surface albedo is set to 90% and solar zenith angle is set to $0°$." was deleted.

**GC 2-2**

Retrieval error estimations show extremely high levels (i.e. PCE from Figure 3). These values (around 65%) are too high for a proper retrieval. This is the case for all retrievals using diverse spectral domains (even for single band retrievals). We expect typical retrieval errors for tropospheric ozone around 20 to 30% (IR, UV, etc) and NOT 65%. Retrieval errors of diverse tropospheric ozone columns for most single band retrievals and even UV+IR approaches implemented to real data are indeed of 20-30%. This aspect should be revised in the simulations presented by Kasai et al.. For example, retrievals can be more constrained for obtaining lower errors and therefore will provide lower sensitivity. Sensitivity is then overestimated when accepting too high errors. Therefore, the retrieval approach used in the paper to estimate averaging kernels and errors in the results presented in the paper (Figures 3, 4 and 5) should be performed again.

**Answer to GC 2-2**

We thank the referee for giving us this valuable comment and drawing our attention to this point. We assumed 100% a priori error in our simulation to clearly see the rate of the error reduction by the measurement. This is the reason of high partial column error (PCE) such as 65%. No feasibility studies of adding MW limb measurement to the ozone synergetic retrieval approach have been reported as far as we know. Our study is the first attempt. We believe that a feasibility study in an ideal condition is quite important as the first step to introduce the MW limb measurement into the synergy retrieval of the tropospheric ozone. The values of PCE were approximately 55-65%, so that 35-45% error was reduced by the measurement spectra.

But as you pointed it out, 65% error is higher than typical retrieval error of 20-30% and showing the 65% error might be misleading for readers. Therefore, we employed the reduction rate of error (RRE) for the partial column as an alternative of PCE. The RRE value was given by

$$RRE = \frac{PCE_{apriori} - PCE_{retrieved}}{PC},$$

where $PCE_{apriori}$ is PCE for the a priori state, $PCE_{retrieved}$ is PCE for the retrieved state, and PC is the partial column of ozone. The manuscript was improved as follows.

**Revisions to GC 2-2**

Page 8 Line 28 – Page 9 Line 3

"The diagonal components of $S_a$ were the squares of the a priori error $\sigma_a$ (100% of the log-based VMR) at each vertical pressure grid)."

$\rightarrow$

"The diagonal components of $S_a$ were the squares of the a priori error $\sigma_a$ at each vertical pressure grid). The value of $\sigma_a$ was set to 100% of the log-based a priori VMR to simply quantify the error reduction from the error in the a priori error to the error in the retrieved state due to the measurement."

Page 9 Line 24 – Page 10 Line 9

"The ozone partial column (PC) for the same vertical layers is defined as

$$PC = \sum_{i=i_{min}}^{i_{max}} \frac{p[i] \cdot VMR[i]}{k_B \cdot T[i]} \Delta z[i]. \qquad (14)$$

Here, p[i], VMR[i], T[i] and $\Delta z[i]$ are pressure, VMR of ozone, temperature, and the vertical length of the $i$th layer, respectively. $k_B$ is the Boltzmann constant. We defined partial column error, PCE, as the relative error in PC by

$$PCE = \frac{1}{PC} \sum_{i=i_{min}}^{i_{max}} \frac{p[i] \cdot \varepsilon_{VMR}[i]}{k_B \cdot T[i]} \Delta z[i]. \qquad (15)$$

$\varepsilon_{VMR}[i]$ is the total retrieval error in ozone VMR at the $i$th layer."

$\rightarrow$

"The RRE is given by

$$RRE = \frac{PCE_{apriori} - PCE_{retrieved}}{PC}, \qquad (14)$$

where $PCE_{apriori}$ and $PCE_{retrieved}$ are the partial column error, PCE, for the a priori state and the retrieved state, respectively. PC represents the partial column of ozone and the value of PC from the $i_{min}$th vertical layer to the $i_{max}$th layer is given by

$$PC = \sum_{i=i_{min}}^{i_{max}} \frac{p[i] \cdot VMR[i]}{k_B \cdot T[i]} \Delta z[i]. \qquad (15)$$

Here, $p[i]$, $VMR[i]$, $T[i]$ and $\Delta z[i]$ are pressure, VMR of ozone, temperature, and the vertical length of the $i$th layer, respectively. $k_B$ is the Boltzmann constant. The value of PCE is given by

$$PCE = \sum_{i=i_{min}}^{i_{max}} \frac{p[i] \cdot \varepsilon_{VMR}[i]}{k_B \cdot T[i]} \Delta z[i]. \qquad (16)$$

$\varepsilon_{VMR}[i]$ is the total retrieval error in ozone VMR at the $i$th layer ($\sigma_a$ for $PCE_{apriori}$ and $\varepsilon_x$ for $PCE_{retrieved}$)."

Page 11 Lines 24 – 29

"The relative error in the partial column of ozone, PCE, calculated using Eq. (15), is shown in the right column of Fig. 3. PCE generally decreased when more wavelength regions were used. PCE was approximately 55-70% in the LMT, MT and UT regions. PCE for the MW measurements alone could not be estimated in the MT and LMT because the DFS values were almost zero and there was no sensitivity with which to retrieve the ozone amount from the MW measurements."
→

"The reduction rate of error, RRE, in the ozone partial column of ozone calculated using Eq. (14), is shown in the right column of Fig. 4. The value of RRE was approximately 25-45% in the UT, MT and LMT regions. The RRE generally increased by combining more wavelength ranges in the ozone synergetic retrieval as DFS shown in Fig. 3. The RRE value for all profiles averaged in the LMT region was 36% and 39% for UV+TIR and UV+TIR+MW measurements, respectively. Adding the MW measurement made 3% increase of RRE value. A certain increase of the retrieval sensitivity of the LMT ozone was shown by DFS as well as PMS and RRE."

**GC 2-3**

Explicit comparison with other multispectral synergisms: The added value of the method proposed in the paper should gain great clarity with an explicit comparison with previous work on multispectral retrieval of ozone. The performance of the UV+IR+MW synergism (sensitivity and probed height) and characteristics of the instruments (spectral resolution, signal-to-noise ratio) considered in the paper should be compared to previous multispectral approaches to retrieve ozone from space. This comparison should be done with the multispectral methods implemented with real measurements (OMI and TES, Fu et al., 2013 and GOME-2 and IASI, Cuesta et al. 2013) and also future approaches (e.g. UVNS and IASI-NG, Costantino et al., 2017).

**Answer to GC 2-3**

We deeply appreciate your valuable comment. As you mentioned, our calculation results should be validated with a comparison with previous studies of the synergetic retrieval of the tropospheric ozone. The DFS values themselves vary in the simulation situations and instrumental characteristics, so we compared a relative difference between the mean of the DFS values of UV only and TIR only and the DFS value of the UV+TIR measurements. The DFS relative difference was compared with those reported from Fu et al., (2013) and Cuesta et al., (2013). We focused on the LMT region because of our most interest.

In our simulation, the DFS values for UV only and TIR only were 0.195 and 0.210, respectively, for all profiles averaged. Their mean value is 0.203. The DFS value for the UV+TIR measurement was 0.458. The relative difference was estimated to be 126% (=(0.458-0.203)/0.203). The same calculation was applied to the DFS values of Fu et al., (2013) and Cuesta et al., (2013) as follows. Fu et al., (2013) showed that the DFS values for the UV (OMI) and TIR (TES) were 0.10 and 0.21, respectively. Their mean value was 0.155. The DFS for UV+TIR (OMI+TES) was 0.37. The relative difference was estimated to be 139% (=(0.37-0.155)/0.155). In the case of Cuesta et al., (2013), the DFS values over land and ocean were averaged. The DFS value of UV (GOME-2) was 0.08 for both land and ocean, and their mean value was also 0.08. In the same way, the DFS values of TIR (IASI) were 0.24 and 0.16 for land and ocean, respectively, and their mean was 0.20. The mean of the DFS values for UV and TIR was 0.14. The DFS values of UV+TIR (GOME-2+IASI) were 0.34 and 0.23 for land and ocean, respectively, and their mean was 0.285. Therefore, the relative difference between the mean of the DFS values of UV only and TIR only and the DFS value of the UV+TIR measurements was calculated to be 104% (=(0.285-0.14)/0.14). The DFS relative difference in our simulation (126%) showed good agreement with those in the previous studies and was between the two (139% and 104%). The simulation results performed by Natraj et al., (2011) was also compared. We summarized these comparisons in one Table, and added the description in out manuscript as follows.

Costantino et al., (2017) performed multispectral synergy retrieval for the future missions of IASI-NG and UVNS. They discussed increase of the sensitivity of the tropospheric ozone retrieval by improvement of the instrument rather than combining the UV and TIR measurements, and showed that the DFS values of combination of IASI-NG and UVNS were 0.75 and 0.66 over land and ocean, respectively. Our simulation showed approximately 30% increase of DFS value in the LMT. We estimated the DFS values in the LMT ozone by adding the MW limb measurement to the IASI-NG+UVNS synergetic retrieval to be 0.98 (=0.75×1.3) and 0.86 (=0.66×1.3) for land and ocean, respectively. We added this estimation as the perspective of the future achievement of the ozone synergetic approach.

Page 9 Line 24 – Page 10 Line 1

"We evaluated the sensitivity of ozone retrieval for seven wavelength combinations in terms of the DFS and partial column error (PCE). We calculated DFS and PCE for the UT, MT and LMT regions as follows. The value of DFS from the $i_{min}$th vertical layer to the $i_{max}$th layer is given by

$$DFS = \sum_{i=i_{min}}^{i_{max}} A[i, i] \qquad (13)"$$

$\rightarrow$

"We evaluated the sensitivity of ozone retrieval for seven wavelength combinations in terms of DFS. We calculated the DFS values for the partial column in the UT, MT and LMT regions. The value of DFS from the $i_{min}$th vertical layer to the $i_{max}$th layer is given by

$$DFS = \sum_{i=i_{min}}^{i_{max}} A[i, i]. \qquad (13)"$$

We also evaluated the sensitivity of ozone retrieval using the pressure of maximum sensitivity (PMS) and the reduction rate of error (RRE) for the partial column. The PMS was defined as the pressure of the maximum of the sum of rows of the corresponding A for the ozone partial column."

Page 10 Line 12

"The left column of Fig. 3 shows" $\rightarrow$ "Figure 3 and Table 3 show"

Page 11 Lines 17 – 23

"The pressure of maximum sensitivity, PMS, for the ozone partial column should be located in a range of the corresponding partial column. In the UT region, the PMS values for all cases were located in a range of the corresponding region (215-383 hPa) by combining more than two wavelength ranges. It was also observed in the PMS values in the MT region. But in the LMT region, only the PMS values of combination of the three wavelength ranges were located in the vertical region of LMT. The PMS value for all profiles averaged in the LMT region was 783 hPa and 808 hPa for the UV+TIR and UV+TIR+MW measurements, respectively. The PMS value was increased by approximately 3% by adding the MW measurement to the UV+TIR measurements, although the PMS value of the MW measurement itself was lower than 300 hPa in the LMT region." was added.

Pages 21 – 22 Figures 3 and 4

[Figure]

Figure 3. Values of DFS and PCE in the upper troposphere (UT, 215-383 hPa), middle troposphere (MT, 383-749 hPa), and lowermost troposphere (LMT, >749 hPa): (red) June 2009 in CEC, (purple) June 2009 in ECS, (green) December 2009 in CEC, (blue) December 2009 in ECS and (black) all of 20 profiles.

→

[Figure]

Figure 3. Values of DFS in the upper troposphere (UT, 215-383 hPa), middle troposphere (MT, 383-749 hPa), and lowermost troposphere (LMT, >749 hPa): (red) June 2009 in CEC, (purple) June 2009 in ECS, (green) December 2009 in CEC, (blue) December 2009 in ECS and (black) all of 20 profiles.

[Figure]

Figure 4. Same as Fig. 3 but for the PMS (left) and RRE (right). The gray shaded area represents the vertical region that corresponds to UT, MT and LMT.

Pages 23 – 24 Figures 5 and 6

The figure numbers of the averaging kernel plots were renumbered to Figures 5 and 6.

**GC 2-5**

Additional terms for error estimations: for such a new multispectral retrieval, I strongly recommend to estimate the contribution of additional terms of errors such as cross errors from joint fit of surface albedo and emissivity and other systematic errors (atmospheric and surface temperature, water vapor, etc).

**Answer to GC 2-5**

We deeply appreciate the referee for pointing it out. The main purpose of this feasibility study is evaluation of adding the MW limb measurement to the synergetic retrieval of the tropospheric ozone. We added the description of large error sources in the MW limb measurement of spectroscopic parameters. Time difference between the MW limb measurement and UV and TIR nadir measurements was also discussed as follows.

**Revisions to GC 2-5**

Page 12 Lines 16 – 30

"In this study, we showed that an introduction of the MW limb measurement had a certain effect

to increase the sensitivity of the tropospheric ozone retrieval. However, following issues might cause bias and uncertainties in the retrieval results and should be considered to implement this retrieval method to real measurements. Discrepancy in spectroscopic parameters for several wavelength ranges is one of the most important error sources. For the ozone retrieval using the MW limb measurement, spectroscopic parameters are the largest error sources. It was reported that approximately 3-5% error was caused by uncertainties in air-broadening coefficient and line intensity in the case of the SMILES observation (Kasai et al., 2013). It is comparable to the approximately 4% uncertainty in the spectroscopic parameters in the UV and TIR wavelength ranges (Gratien et al., 2010). The tangent height correction can also be a large error source for the MW limb measurement (Kasai et al., 2013). The tangent height is a key parameter to determine the field-of-view, thus uncertainty in the tangent height causes discrepancy of the atmospheric layer assumed in the simulation and the true atmospheric layer. This discrepancy critically affects to the retrieval of the ozone amount in both the stratosphere and the troposphere, and also might cause bias for correction of time delay between the MW limb measurement and the other nadir measurements. In this study, we assumed instruments onboard the ISS (low orbit) and the time difference of approximately five minutes could be ignored. If the time difference was long and its correction was required, three-dimensional atmospheric modeling should be performed including the field-of-view of the MW limb measurement." was added.

**GC 2-6**

A better description of the atmospheric scenario is needed. How model and airborne data are combined? How the 20 cases are chosen? What is the main characteristic (time of the day, region, ozone load, surface and atmospheric properties) of each scenario? These aspects should be explicitly commented and described in the text of the paper.

**Answer to GC 2-6**

We deeply appreciate this suggestion. We synthesized the vertical profiles of ozone, water vapor and temperature from three atmospheric profiles; a one-way nested global-regional air quality forecasting (AQF) system [Takigawa et al., 2007, 2009], the Modern Era Retrospective-Analysis for Research and Applications (MERRA) data [Rienecker et al., 2011] and the COSPAR International Reference Atmosphere (CIRA) [Fleming et al., 1990]. The aircraft observation data was used in the MERRA data, but we did not use the aircraft observation data directly. The statement "We made a total of 20 atmospheric scenarios from the model calculation and air-born observations made over two Asian areas (CEC and ECS) in June and December 2009. (Page 4 Lines 32 – 33)" was quite misleading, and we improved this statement as follows.

The reason why we chose the 20 scenarios over Asian area, CEC and ECS, was that China is one of the largest air-polluted countries in the world and the prediction of transboundary pollution is one of the main purposes of the Japanese future missions. The CEC, located between Beijing and Shanghai, is the area where largest amount of ozone in the lowermost troposphere was observed over East Asia from the Aura/OMI measurement [Hayashida et al., 2015]. This work also showed the amount of ozone in the lowermost troposphere in the urban area was largest in June (summer season) and smallest in December (winter season). The ECS was selected for a comparison between the urban area (CEC) and ocean (ECS). The main characteristics of the 20 scenarios are shown in Table 2, and we added the explanation of the 20 scenarios in our manuscript as follows.

[revised manuscript text omitted]

**GC 2-7**

Spectroscopy coherence should be mentioned as an important issue for real retrievals. There exist previous multispectral retrievals combining microwave spectra with other domains? Spectroscopic consistency has been analyzed as for UV and IR (e.g. Gratien et al., 2010)?

**Answer to GC 2-7**

We thank the referee for drawing our attention to this point. As far as we know, our study is the first attempt to introduce the MW measurement to the synergetic retrieval of the tropospheric ozone. We referred Gratien et al., (2010) and discussed the tropospheric ozone retrieval error due to uncertainties in the spectroscopic parameters in the MW ranges as follows.

"from 1.00 in the TIR+UV measurements to 1.23 (about a 20% increase)"

→

"from 1.00 to 1.23 (about 23% increase) for the TIR+UV measurements"

Page 11 Line 1

"in the MT and LT" → "in the UT and MT"

Page 11 Line 5

"a 40%" → "30%"

Page 12 Line 2

[revised manuscript text omitted]
 ($30^\circ$N–$40^\circ$N, $110^\circ$E–$123^\circ$E) and the East China Sea ($29^\circ$N–$33^\circ$N, $125^\circ$E–$129.5^\circ$E), respectively.

[c] Temperature and pressure at the surface.

[d] PC means ozone partial column. PC is presented for each altitude region: upper troposphere (UT, 215–383 hPa), middle troposphere (MT, 383–749 hPa), and lowermost troposphere (LMT, $>$ 749 hPa).

[e] $H_2O$ column amount in the troposphere.

**Table 3.** Mean and standard deviation (1-$\sigma$) of DFS in the upper troposphere (UT, 215–383 hPa), middle troposphere (MT, 383–749 hPa), and lowermost troposphere (LMT, >749 hPa). The standard deviation is listed in the parenthesis. For example, 0.16(6) means 0.16 of the mean value and 0.06 of the standard deviation.

|  |  | UV | TIR | MW | TIR+MW | UV+MW | UV+TIR | UV+TIR+MW |
|---|---|---|---|---|---|---|---|---|
| 6, 2009/CEC | UT | 0.16(6) | 0.53(4) | 0.36(20) | 1.16(14) | 0.91(20) | 0.55(4) | 1.17(14) |
|  | MT | 0.61(3) | 0.94(2) | <0.01 | 1.08(4) | 0.78(3) | 1.00(2) | 1.17(1) |
|  | LMT | 0.37(6) | 0.44(6) | <0.01 | 0.48(6) | 0.48(5) | 0.87(3) | 1.03(1) |
| 6, 2009/ECS | UT | 0.15(7) | 0.56(10) | 0.03(3) | 0.89(9) | 0.60(10) | 0.57(8) | 0.92(10) |
|  | MT | 0.61(11) | 0.88(12) | <0.01 | 0.99(11) | 0.74(13) | 0.99(13) | 1.11(13) |
|  | LMT | 0.23(13) | 0.17(11) | <0.01 | 0.18(11) | 0.30(14) | 0.56(15) | 0.70(16) |
| 12, 2009/CEC | UT | 0.12(3) | 0.65(8) | 0.72(17) | 1.34(8) | 1.06(12) | 0.68(1) | 1.40(7) |
|  | MT | 0.34(9) | 0.75(4) | <0.01 | 1.04(3) | 0.69(2) | 0.98(3) | 1.33(2) |
|  | LMT | 0.08(2) | 0.09(3) | <0.01 | 0.14(1) | 0.12(2) | 0.19(1) | 0.33(3) |
| 12, 2009/ECS | UT | 0.23(8) | 0.63(8) | 0.80(40) | 1.35(23) | 1.19(30) | 0.66(4) | 1.47(27) |
|  | MT | 0.46(9) | 0.77(5) | <0.01 | 1.04(9) | 0.72(5) | 1.05(5) | 1.31(7) |
|  | LMT | 0.18(2) | 0.30(9) | <0.01 | 0.13(1) | 0.25(1) | 0.36(2) | 0.48(5) |
| All profiles | UT | 0.16(7) | 0.59(9) | 0.44(40) | 1.16(25) | 0.90(30) | 0.62(8) | 1.21(28) |
|  | MT | 0.50(15) | 0.83(11) | <0.01 | 1.03(8) | 0.73(9) | 1.00(9) | 1.23(13) |
|  | LMT | 0.20(12) | 0.21(15) | <0.01 | 0.20(14) | 0.26(15) | 0.46(25) | 0.59(26) |

**Table 4.** Comparison of DFS in the LMT region with previous studies. The method and situation to derive the DFS value are different for each study, thus, the relative difference between the DFS value for the UV+TIR measurement ($DFS_{UV+TIR}$) and mean of DFS values for UV and TIR measurements ($DFS_{UV}$, $DFS_{TIR}$) was used for this comparison. The relative difference was calculated by ($DFS_{UV+TIR}$ − Mean($DFS_{UV}$, $DFS_{TIR}$))/Mean($DFS_{UV}$, $DFS_{TIR}$).

| | $DFS_{UV}$ | $DFS_{TIR}$ | Mean($DFS_{UV}$, $DFS_{TIR}$) | $DFS_{UV+TIR}$ | Rel. dif. | Definition of LMT |
|---|---|---|---|---|---|---|
| This work | 0.20 | 0.21 | 0.203 | 0.458 | 126 % | >749 hPa |
| Fu et al. (2013) | 0.10 | 0.21 | 0.15 | 0.37 | 139 % | >700 hPa |
| Cuesta et al. (2013)[a] | 0.08 | 0.20 | 0.14 | 0.285 | 104 % | <3 km |
| Natraj et al. (2011) | 0.26 | 0.27 | 0.265 | 0.57 | 115 % | >800 hPa |

[a] DFS values over land and ocean are averaged.

---

## Author Response (AR1)

**Reply to Referee #1**

Dear Referee #1

We deeply appreciate to Referee #1 for valuable comments and suggestions. Please find the manuscript with major revision. Answered to your comments/questions are described in point by point. We also improved the introduction to address the main purpose of this paper. We hope that current manuscript is significant for the publication in AMT. We still wonder our English is not perfect as a manuscript of AMT even after the Native English checking. We will ask to our colleague, who is native English speaker, only once before the final version, since we do not want to disturb him frequently.

Sincerely yours,

Tomohiro Sato and Yasuko Kasai
National Institute of Information and Communications Technology

Note
**GC**: General comments
**MC**: Minor corrections or Additional corrections

**GC 1-1**

This paper presents the results of a study designed to facilitate the combination of space borne spectral measurements in the ultraviolet, thermal infrared and microwave wavelength ranges to determine ozone concentration in the tropospheric/lower stratosphere. The study combines nadir viewing (UV and TIR) with limb-sounding (MV) remote sounding techniques. Standard optimal estimation retrieval theory is used to evaluate the expected performance of this triad of remote sounding measurements. Results presented include degrees of freedom for signal, partial column errors and elements of the averaging kernel matrices for a range of East Asian summer and winter atmospheric conditions. Although the MW instrument itself has no sensitivity below the middle troposphere, the MW provides additional constraints in the stratosphere that actually benefit the retrievals of ozone in the lowermost troposphere than compared to using UV+TIR alone.

The paper is well written and the material is presented in a clear manner. My main concern though is the extent to which the study is applicable to real world instruments, since for example the complications arising from clouds are not considered and neither are the inevitable inconsistencies of instrument biases, both which of course are the bane of researchers trying to perform multi-instrument simultaneous retrievals. I would have liked at least some discussion on these points. Aerosols are mentioned in connection with the UV instrument, but it is not clear if these are in a layer below the sensitivity of the TIR instrument.

I recommend publication of this paper in AMT.

**Answer to GC 1-1**

We appreciate your valuable comments and suggestions. As you mentioned, discussion of error sources for the tropospheric ozone retrieval is quite valuable because there are many problems to overcome to realize the multi-instrument simultaneous retrieval using real measurements. Our major aim of this feasibility study is to evaluate the sensitivity of the tropospheric ozone retrieval adding the MW measurement. We added the discussion of error sources in the MW limb measurement (spectroscopic parameters and correction of field-of-view) and expected problems to perform the synergetic retrieval with the MW limb measurement and UV, TIR nadir measurement (time difference between the limb and nadir measurements).

Aerosol profile was included in the UV simulation as a known parameter. We did not include the aerosol profile in the TIR simulation because of the following reason. The extinction of radiation due to aerosol particle goes down in a linear to quadratic fashion with wavelength. The wavelength of TIR range corresponds to about 9.6 μm and the extinction of radiation due to such a large aerosol particle is negligibly small (e.g., Section 7.4 of Natraj et al., 2011). If the effect of aerosol could not be ignored in the TIR range, then it is the case as quite large enhancement of

yellow dust on the desert. As the first feasibility study of introducing the MW limb measurement into the synergetic retrieval of tropospheric ozone, simulation of the ozone synergetic retrieval in such a rare case is beyond the scope of this paper.

We also improved the introduction to clearly address the aim of this paper. We hope that our manuscript is much improved and suitable for publication in AMT.

References
- Natraj, V., Liu, X., Kulawik, S., Chance, K., Chatfield, R., Edwards, D.~P., Eldering, A., Francis, G., Kurosu, T., Pickering, K., Spurr, R. and Worden, H.: Multi-spectral sensitivity studies for the retrieval of tropospheric and lowermost tropospheric ozone from simulated clear-sky GEO-CAPE measurements, Atmospheric Environment, 45, 7151-7165, 2011.

**Revisions to GC 1-1**
Page 3 Lines 29 – 32
"Our major aim of this feasibility study is to evaluate the sensitivity of the tropospheric ozone retrieval adding the MW measurement to the multi-spectral synergetic retrieval. Thus, the feasibility study was performed under an ideal condition for the synergetic retrieval of the tropospheric ozone." was added.

Page 6 Line 32 – Page 7 Line 1
"The aerosol profile was not included in the TIR calculation. The extinction of radiation due to aerosol particles with a scale of approximately 9.6 μm, which corresponds to the wavelength of the TIR range, is negligibly small for the synergetic retrieval of the LMT ozone with the TIR measurement (e.g., Natraj et al., 2011)." was added.

Page 12 Lines 16 – 30
"In this study, we showed that an introduction of the MW limb measurement had a certain effect to increase the sensitivity of the tropospheric ozone retrieval. However, following issues might cause bias and uncertainties in the retrieval results and should be considered to implement this retrieval method to real measurements. Discrepancy in spectroscopic parameters for several wavelength ranges is one of the most important error sources. For the ozone retrieval using the MW limb measurement, spectroscopic parameters are the largest error sources. It was reported that approximately 3-5% error was caused by uncertainties in air-broadening coefficient and line intensity in the case of the SMILES observation (Kasai et al., 2013). It is comparable to the approximately 4% uncertainty in the spectroscopic parameters in the UV and TIR wavelength ranges (Gratien et al., 2010). The tangent height correction can also be a large error source for the

MW limb measurement (Kasai et al., 2013). The tangent height is a key parameter to determine the field-of-view, thus uncertainty in the tangent height causes discrepancy of the atmospheric layer assumed in the simulation and the true atmospheric layer. This discrepancy critically affects to the retrieval of the ozone amount in both the stratosphere and the troposphere, and also might cause bias for correction of time delay between the MW limb measurement and the other nadir measurements. In this study, we assumed instruments onboard the ISS (low orbit) and the time difference of approximately five minutes could be ignored. If the time difference was long and its correction was required, three-dimensional atmospheric modeling should be performed including the field-of-view of the MW limb measurement." was added.

**MC 1-1**

P4,L20: MLS also has other ozone bands (in the 190GHz, 640GHZ and THz radiometers) in addition to the standard product at 240GHz.

**Answer to MC 1-1**

Thank you so much for pointing it out. We improved the statement as follows. (We did not include the transition in THz range because the THz range is not the wavelength range of our calculation.)

**Revisions to MC 1-1**

Page 4 Line 34 – Page 5 Line 1

"e.g., 243.4 GHz for Aura/MLS (Waters et al., 2006), 501.8 and 544.6 GHz for Odin/SMR (Urban et al., 2005), and 625.3 GHz for JEM/SMILES (Kikuchi et al., 2010)"

→

"e.g., 206.1 and 235.7 GHz for Aura/MLS (Waters et al., 2006), 501.8 and 544.6 GHz for Odin/SMR (Urban et al., 2005), and 625.4 GHz for Aura/MLS and JEM/SMILES (Kikuchi et al., 2010)"

**MC 1-2**

P4,L30: This seems to be a suggesting 17.5 second vertical limb scan. It would be worthwhile to point this out in connection with Fig 1. Also the paper state that the time delay between nadir and limb views (5 mins) has been ignored. Is that because the time difference has negligible effect on the atmospheric scenes?

**Answer to MC 1-2**

Yes, we assumed 17.5 seconds vertical limb scan. The vertical range of limb scan is 10 – 80 km with an interval of 2 km, thus, total 35 spectra are acquired in one vertical limb scan. An integration time to acquire one spectrum is 0.5 seconds, and it takes 17.5 seconds for one vertical limb scan. We added this information in the manuscript as follows.

Also, we ignored the time delay between the nadir and limb measurements because of the following reason. The platform of the three instruments in this simulation study is assumed to be the ISS. The height and speed of the ISS is assumed to be 300 km and 7.8 km/s. In this case, the time delay between the coincidence of limb and nadir observation is calculated to be approximately 5 minutes. Assuming a typical horizontal wind speed in the troposphere (e.g., ~1.2 km/min [Fleming et al., 1988]), we consider that the horizontal displacement of the air within 5 minutes will be approximately ~6 km. Such a displacement is within a typical horizontal resolution of limb measurement, 8 km (Aura/MLS) and 20 km (Aura/TES). Therefore, we ignored any effects from the time delay between the nadir and limb measurements.

Reference
- Fleming, E. L., S. Chandra, M. R. Schoeberl, and J. J. Barnett, "Monthly mean global climatology of temperature, wind, geopotential height, and pressure for 0 – 120 km", NASA Tech Memo 100687, 85 pp., 1988.

**Revisions to MC 1-2**

Page 5 Lines 7 – 12

"The Earth's limb was assumed to be scanned vertically from 10 to 80 km with an interval of 2 km using a 40 cm diameter antenna. The brightness temperature noise was estimated to be 0.7 K and 1.7 K for the 350 GHz and 645 GHz bands, respectively, assuming the system noise temperature of a typical Shottky-barrier mixer (2500 K and 6000 K for 350 GHz band and 645 GHz band, respectively) and a typical integration time of 0.5 s for one spectrum accumulation."

→

"The antenna diameter was assumed to be 40 cm. The Earth's limb was assumed to be scanned vertically from 10 to 80 km with an interval of 2 km and total 35 spectra were acquired in one vertical limb scan. We also assumed that a typical integration time was 0.5 seconds for one spectrum accumulation and it took 17.5 seconds for one vertical limb scan. The brightness temperature noise was estimated to be 0.7 K and 1.7 K for the 350 GHz and 645 GHz bands, respectively, assuming the system noise temperature of a typical Shottky-barrier mixer (2500 K and 6000 K for 350 GHz band and 645 GHz band, respectively)."

Page 4 Lines 10 – 13

"We ignored this time difference between the UV and TIR measurements and MW measurement."

→

"The time delay of 5 minutes corresponds to approximately 6 km transport when we assume a typical value of horizontal wind speed in the troposphere and the stratosphere of 1.2 km/min (Fleming et al., 1988), which is smaller than typical horizontal resolution of the MW limb measurement (8 km for Aura/MLS (Waters et al., 2006)). Therefore, we ignored this time difference between the UV and TIR measurements and MW measurement."

Reference

"Fleming, E. L., S. Chandra, M. R. Schoeberl, and J. J. Barnett, "Monthly mean global climatology of temperature, wind, geopotential height, and pressure for 0 – 120 km", NASA Tech Memo 100687, 85, 1988." was added.

**MC 1-3**

P7,L28: There is almost certainly covariance in the apriori, but you are using none. Have you done any simulations with off-diagonal components in Sa to see how the results are affected?

**Answer to MC 1-3**

We thank the referee for pointing it out. As you mentioned, there is certain correlations between the ozone concentrations in different vertical layers. But one of the aims of this paper is to investigate which wavelength range, UV, TIR and MW, contributes to the ozone retrieval in which vertical region of the upper, middle and lowermost troposphere. If we set certain values in off-diagonal components in $S_a$, such a correlation in a priori vertical profile assists the retrieval of ozone in a layer even in which ozone information is not sufficiently included in a measurement spectrum (e.g., Saitoh et al., 2009). This may obscure what we want to investigate in this feasibility study. We set off-diagonal components in $S_a$ to be zero to avoid the assistance of the ozone retrieval by the correlations between different vertical layers. We added the statement to explain the reason why the off-diagonal components in $S_a$ are set to zero as follows.

Reference

- Saitoh, N., Imasu, R., Ota, Y. and Niwa, Y.: $CO_2$ retrieval algorithm for the thermal infrared spectra of the Greenhouse Gases Observing Satellite: Potential of retrieving $CO_2$ vertical profile from high-resolution FTS sensor, Journal of Geophysical Research, 114, D17305, doi:10.1029/2008JD011500. 2009.

**Revisions to MC 1-3**

Page 9 Lines 3 – 10

"The off-diagonal components of $S_a$ and $S_\varepsilon$ were set to zero."

$\rightarrow$

"The off-diagonal components of $S_\varepsilon$ were set to zero. The off-diagonal components in $S_a$ indicate the correlations between the ozone concentrations in different vertical layers. Non-zero off-diagonal components in $S_a$ assist the retrieval of ozone concentration in a layer in which sufficient ozone information is not included in a measurement spectrum with the correlations with other layers in which sufficient ozone information is included in a measurement spectrum (e.g., Saitoh et al., 2009). One of the aims of our feasibility study is to investigate the ozone retrieval sensitivity for each vertical region in an ideal condition. We set off-diagonal components in $S_a$ to be zero to avoid the assistance of the ozone retrieval by the correlations between different vertical layers."

**MC 1-4**

P9: A lot of values are given in the text. Could these be placed in a table?

**Answer to MC 1-4**

We thank the referee for pointing it out. We discussed the ozone retrieval sensitivity with DFS values, and a lot of DFS values were stated. As you suggested, it could be better to show the DFS values in one table. We added the table of the DFS values as follows (Table 3).

**Revisions to MC 1-4**

Page 10 Line 12

"The left column of Fig. 3 shows" $\rightarrow$ "Figure 3 and Table 3 show"

P 27 Table 3 was added.

Table 3. Mean and standard deviation (1-σ) of DFS in the upper troposphere (UT, 215-383 hPa), middle troposphere (MT, 383-749 hPa), and lowermost troposphere (LMT, >749 hPa). The standard deviation is listed in the parenthesis. For example, 0.16(6) means 0.16 of the mean value and 0.06 of the standard deviation.

|  |  | UV | TIR | MW | TIR+MW | UV+MW | UV+TIR | UV+TIR+MW |
|---|---|---|---|---|---|---|---|---|
| 6, 2009/CEC | UT | 0.16(6) | 0.53(4) | 0.36(20) | 1.16(14) | 0.91(20) | 0.55(4) | 1.17(14) |
|  | MT | 0.61(3) | 0.94(2) | <0.01 | 1.08(4) | 0.78(3) | 1.00(2) | 1.17(1) |
|  | LMT | 0.37(6) | 0.44(6) | <0.01 | 0.48(6) | 0.48(5) | 0.87(3) | 1.03(1) |
| 6, 2009/ECS | UT | 0.15(7) | 0.56(10) | 0.03(3) | 0.89(9) | 0.60(10) | 0.57(8) | 0.92(10) |
|  | MT | 0.61(11) | 0.88(12) | <0.01 | 0.99(11) | 0.74(13) | 0.99(13) | 1.11(13) |
|  | LMT | 0.23(13) | 0.17(11) | <0.01 | 0.18(11) | 0.30(14) | 0.56(15) | 0.70(16) |
| 12, 2009/CEC | UT | 0.12(3) | 0.65(8) | 0.72(17) | 1.34(8) | 1.06(12) | 0.68(1) | 1.40(7) |
|  | MT | 0.34(9) | 0.75(4) | <0.01 | 1.04(3) | 0.69(2) | 0.98(3) | 1.33(2) |
|  | LMT | 0.08(2) | 0.09(3) | <0.01 | 0.14(1) | 0.12(2) | 0.19(1) | 0.33(3) |
| 12, 2009/ECS | UT | 0.23(8) | 0.63(8) | 0.80(40) | 1.35(23) | 1.19(30) | 0.66(4) | 1.47(27) |
|  | MT | 0.46(9) | 0.77(5) | <0.01 | 1.04(9) | 0.72(5) | 1.05(5) | 1.31(7) |
|  | LMT | 0.18(2) | 0.30(9) | <0.01 | 0.13(1) | 0.25(1) | 0.36(2) | 0.48(5) |
| All profiles | UT | 0.16(7) | 0.59(9) | 0.44(40) | 1.16(25) | 0.90(30) | 0.62(8) | 1.21(28) |
|  | MT | 0.50(15) | 0.83(11) | <0.01 | 1.03(8) | 0.73(9) | 1.00(9) | 1.23(13) |
|  | LMT | 0.20(12) | 0.21(15) | <0.01 | 0.20(14) | 0.26(15) | 0.46(25) | 0.59(26) |

**Other corrections by ourselves**

Page 10 Lines 28 – 19

"from 1.00 in the TIR+UV measurements to 1.23 (about a 20% increase)"

→

"from 1.00 to 1.23 (about 23% increase) for the TIR+UV measurements"

Page 11 Line 1

"in the MT and LT" → "in the UT and MT"

Page 11 Line 5

"a 40%" → "30%"

Page 12 Line 2

"This shows that retrieval combining UV, TIR and MW measurements can possibly retrieve the ozone amount in the LMT region with a DFS value of unity when the LMT ozone is enhanced as large as approximately $5 \times 10^{21}$ m$^{-2}$." was removed.

**Reply to Referee #2**

Dear Referee #2

We would like to thank the Referee #2 for the valuable comments and suggestions. Please find the manuscript with major revision. We answered to your comments and suggestions in point by point. We also improved the abstract and conclusion to make the purpose of this paper clear. We hope that current manuscript is significant for the publication in AMT. We still wonder our English is not perfect as a manuscript of AMT even after the Native English checking. We will ask to our colleague, who is native English speaker, only once before the final version, since we do not want to disturb him frequently.

Sincerely yours,

Tomohiro Sato and Yasuko Kasai
National Institute of Information and Communications Technology

Note
**GC**: General comments
**MC**: Minor corrections or Additional corrections

**Answers to Referee #2**

**Overview comment**

The paper by Kasai et al. presents an interesting simulation study of the potential of a synergetic retrieval using UV, IR and microwave (UV+IR+MW) spectra for retrieving tropospheric ozone vertical profiles. This multispectral approach could be implemented in future Japanese air-quality monitoring missions. Whereas the topic and relevance of the paper are suitable for a possible publication at AMT, the current manuscript needs substantial revision in order to add and complete with important information and correct some inconsistencies of the results. As described below, part of the calculations are needed to be done again (UV radiative transfer and error/averaging kernels estimations) and many aspects should be more detailed (comparisons with previous multispectral methods, analysis of sensitivity as function of height, error estimations, etc.).

The major revisions needed for the manuscript are the following:

**Answer to Overview comment**

We would like to thank you for the helpful comments and suggestions. We carefully checked our calculation script according to your suggestion and found a mistake to estimate the increase ratio of DFS values. The increase ratio was modified to 30% from 40% in the LMT region. Thank you so much for this suggestion. We answered the comments point-by-point and revised according to your suggestions. We also improved the abstract and conclusion in accordance with your suggestions.

**Revisions to Overview comment**

[revised manuscript text omitted]

**GC 2-1**

The choice of some of the parameters used in the simulation are neither clearly justified nor realistic: in page 3 lines 41-43, it is mentioned that a surface albedo of 90% and solar zenith angle of 0° are considered. These parameters are of course not realistic. Indeed, surface albedo at the UV is often from 5% to 15% and only greater than 30% for exceptionally bright surfaces. Solar zenith angles are also key parameters affecting UV spectra that greatly vary depending on overpass time and location of the measurement. Simulations of UV spectra and jacobians should be done again with realistic values of surface albedo and diverse solar zenith angles. The use of appropriate values is key for a proper estimation of the performance of the retrieval approach and they should be realistic for each scene observed.

**Answer to GC 2-1**

We thank the referee for giving us this valuable comment. The statements in our manuscript were misleading, as you mentioned. In our simulation, the surface albedo was assumed to be 0.056 for CEC June, 0.063 for CEC December, 0.065 for ECS June, and 0.084 for ECS December, respectively, as described in Page 7 Lines 5 – 6. These values were obtained from the database that contains the monthly global map of the Earth's surface Lambertian equivalent reflectance (LER) deduced from the Aura/OMI measurements [Kleipool et al., 2008].

The surface albedo value of 90% that you pointed out was used only for estimating the noise equivalent spectral radiance (NESR). The NESR was calculated by dividing the simulated backscattered radiance by the signal-to-noise ratio (SNR). The SNR was estimated by a linear interpolation with the three radiances (the brightest, middle and the smallest) [Private communication with K. Gerilowski]. The smallest radiance was calculated with the condition of 90% albedo and $0^{\circ}$ SZA (cloudy condition). The middle radiance was done in 25% albedo and $45^{\circ}$ SZA. The largest radiance was done in 5% albedo and $80^{\circ}$ SZA. Figure GC2.1.1 shows the results of the SNR simulations. The SNR values were approximately 90 and 1400 (mean of the three spectra) at 305 nm and 340 nm, respectively.

The solar zenith angle (SZA), which corresponds to local time, is also a key parameter for simulation of the UV radiance, as you mentioned. The aim of this article is to investigate the sensitivity of the tropospheric ozone retrieval by adding the MW limb measurement to the synergetic retrieval approach. To purely bring out the improvement of the ozone retrieval sensitivity by combining several wavelength ranges, we performed the simulation in an ideal condition. Landgraf and Hasekamp (2007) showed that the increase of the ozone retrieval sensitivity was largest in the condition of small to moderate SZA (around noon). Therefore, we set the local time to be 11:46am and 00:29pm (around noon) for CEC and ECS, respectively. These local times correspond to the time when the surface ozone is likely increased because the photo-chemistry becomes active and the sensitivity of the ozone retrieval probably becomes high. We thought the local time condition in our simulation was sufficient for investigating the ozone retrieval sensitivity with combination of UV, TIR and MW in an ideal condition. To make it clear, we improved the statements in our manuscript as follows.

Page 8 Line 28 – Page 9 Line 3

"The diagonal components of $S_a$ were the squares of the a priori error $\sigma_a$ (100% of the log-based VMR) at each vertical pressure grid)."

$\rightarrow$

"The diagonal components of $S_a$ were the squares of the a priori error $\sigma_a$ at each vertical pressure grid). The value of $\sigma_a$ was set to 100% of the log-based a priori VMR to simply quantify the error reduction from the error in the a priori error to the error in the retrieved state due to the measurement."

Page 9 Line 24 – Page 10 Line 9

"The ozone partial column (PC) for the same vertical layers is defined as

$$PC = \sum_{i=i_{min}}^{i_{max}} \frac{p[i] \cdot VMR[i]}{k_B \cdot T[i]} \Delta z[i]. \qquad (14)$$

Here, $p[i]$, $VMR[i]$, $T[i]$ and $\Delta z[i]$ are pressure, VMR of ozone, temperature, and the vertical length of the *i*th layer, respectively. $k_B$ is the Boltzmann constant. We defined partial column error, PCE, as the relative error in PC by

$$PCE = \frac{1}{PC} \sum_{i=i_{min}}^{i_{max}} \frac{p[i] \cdot \varepsilon_{VMR}[i]}{k_B \cdot T[i]} \Delta z[i]. \qquad (15)$$

$\varepsilon_{VMR}[i]$ is the total retrieval error in ozone VMR at the *i*th layer."

$\rightarrow$

"The RRE is given by

$$RRE = \frac{PCE_{apriori} - PCE_{retrieved}}{PC}, \qquad (14)$$

where $PCE_{apriori}$ and $PCE_{retrieved}$ are the partial column error, PCE, for the a priori state and the retrieved state, respectively. PC represents the partial column of ozone and the value of PC from the $i_{min}$th vertical layer to the $i_{max}$th layer is given by

$$PC = \sum_{i=i_{min}}^{i_{max}} \frac{p[i] \cdot VMR[i]}{k_B \cdot T[i]} \Delta z[i]. \qquad (15)$$

Here, $p[i]$, $VMR[i]$, $T[i]$ and $\Delta z[i]$ are pressure, VMR of ozone, temperature, and the vertical length of the $i$th layer, respectively. $k_B$ is the Boltzmann constant. The value of PCE is given by

$$PCE = \sum_{i=i_{min}}^{i_{max}} \frac{p[i] \cdot \varepsilon_{VMR}[i]}{k_B \cdot T[i]} \Delta z[i]. \qquad (16)$$

$\varepsilon_{VMR}[i]$ is the total retrieval error in ozone VMR at the $i$th layer ($\sigma_a$ for $PCE_{apriori}$ and $\varepsilon_x$ for $PCE_{retrieved}$)."

Page 11 Lines 24 – 29

"The relative error in the partial column of ozone, PCE, calculated using Eq. (15), is shown in the right column of Fig. 3. PCE generally decreased when more wavelength regions were used. PCE was approximately 55-70% in the LMT, MT and UT regions. PCE for the MW measurements alone could not be estimated in the MT and LMT because the DFS values were almost zero and there was no sensitivity with which to retrieve the ozone amount from the MW measurements."
$\rightarrow$
"The reduction rate of error, RRE, in the ozone partial column of ozone calculated using Eq. (14), is shown in the right column of Fig. 4. The value of RRE was approximately 25-45% in the UT, MT and LMT regions. The RRE generally increased by combining more wavelength ranges in the ozone synergetic retrieval as DFS shown in Fig. 3. The RRE value for all profiles averaged in the LMT region was 36% and 39% for UV+TIR and UV+TIR+MW measurements, respectively. Adding the MW measurement made 3% increase of RRE value. A certain increase of the retrieval sensitivity of the LMT ozone was shown by DFS as well as PMS and RRE."

**GC 2-3**

Explicit comparison with other multispectral synergisms: The added value of the method proposed in the paper should gain great clarity with an explicit comparison with previous work on multispectral retrieval of ozone. The performance of the UV+IR+MW synergism (sensitivity and probed height) and characteristics of the instruments (spectral resolution, signal-to-noise ratio) considered in the paper should be compared to previous multispectral approaches to retrieve ozone from space. This comparison should be done with the multispectral methods implemented with real measurements (OMI and TES, Fu et al., 2013 and GOME-2 and IASI, Cuesta et al. 2013) and also future approaches (e.g. UVNS and IASI-NG, Costantino et al., 2017).

**Answer to GC 2-3**

We deeply appreciate your valuable comment. As you mentioned, our calculation results should be validated with a comparison with previous studies of the synergetic retrieval of the tropospheric ozone. The DFS values themselves vary in the simulation situations and instrumental characteristics, so we compared a relative difference between the mean of the DFS values of UV only and TIR only and the DFS value of the UV+TIR measurements. The DFS relative difference was compared with those reported from Fu et al., (2013) and Cuesta et al., (2013). We focused on the LMT region because of our most interest.

In our simulation, the DFS values for UV only and TIR only were 0.195 and 0.210, respectively, for all profiles averaged. Their mean value is 0.203. The DFS value for the UV+TIR measurement was 0.458. The relative difference was estimated to be 126% (=(0.458-0.203)/0.203). The same calculation was applied to the DFS values of Fu et al., (2013) and Cuesta et al., (2013) as follows. Fu et al., (2013) showed that the DFS values for the UV (OMI) and TIR (TES) were 0.10 and 0.21, respectively. Their mean value was 0.155. The DFS for UV+TIR (OMI+TES) was 0.37. The relative difference was estimated to be 139% (=(0.37-0.155)/0.155). In the case of Cuesta et al., (2013), the DFS values over land and ocean were averaged. The DFS value of UV (GOME-2) was 0.08 for both land and ocean, and their mean value was also 0.08. In the same way, the DFS values of TIR (IASI) were 0.24 and 0.16 for land and ocean, respectively, and their mean was 0.20. The mean of the DFS values for UV and TIR was 0.14. The DFS values of UV+TIR (GOME-2+IASI) were 0.34 and 0.23 for land and ocean, respectively, and their mean was 0.285. Therefore, the relative difference between the mean of the DFS values of UV only and TIR only and the DFS value of the UV+TIR measurements was calculated to be 104% (=(0.285-0.14)/0.14). The DFS relative difference in our simulation (126%) showed good agreement with those in the previous studies and was between the two (139% and 104%). The simulation results performed by Natraj et al., (2011) was also compared. We summarized these comparisons in one Table, and added the description in out manuscript as follows.

Costantino et al., (2017) performed multispectral synergy retrieval for the future missions of IASI-NG and UVNS. They discussed increase of the sensitivity of the tropospheric ozone retrieval by improvement of the instrument rather than combining the UV and TIR measurements, and showed that the DFS values of combination of IASI-NG and UVNS were 0.75 and 0.66 over land and ocean, respectively. Our simulation showed approximately 30% increase of DFS value in the LMT. We estimated the DFS values in the LMT ozone by adding the MW limb measurement to the IASI-NG+UVNS synergetic retrieval to be 0.98 (=0.75×1.3) and 0.86 (=0.66×1.3) for land and ocean, respectively. We added this estimation as the perspective of the future achievement of the ozone synergetic approach.

Page 9 Line 24 – Page 10 Line 1

"We evaluated the sensitivity of ozone retrieval for seven wavelength combinations in terms of the DFS and partial column error (PCE). We calculated DFS and PCE for the UT, MT and LMT regions as follows. The value of DFS from the $i_{min}$th vertical layer to the $i_{max}$th layer is given by

$$DFS = \sum_{i=i_{min}}^{i_{max}} A[i,i] \qquad (13)"$$

$\rightarrow$

"We evaluated the sensitivity of ozone retrieval for seven wavelength combinations in terms of DFS. We calculated the DFS values for the partial column in the UT, MT and LMT regions. The value of DFS from the $i_{min}$th vertical layer to the $i_{max}$th layer is given by

$$DFS = \sum_{i=i_{min}}^{i_{max}} A[i,i]. \qquad (13)"$$

We also evaluated the sensitivity of ozone retrieval using the pressure of maximum sensitivity (PMS) and the reduction rate of error (RRE) for the partial column. The PMS was defined as the pressure of the maximum of the sum of rows of the corresponding A for the ozone partial column."

Page 10 Line 12

"The left column of Fig. 3 shows" $\rightarrow$ "Figure 3 and Table 3 show"

Page 11 Lines 17 – 23

"The pressure of maximum sensitivity, PMS, for the ozone partial column should be located in a range of the corresponding partial column. In the UT region, the PMS values for all cases were located in a range of the corresponding region (215-383 hPa) by combining more than two wavelength ranges. It was also observed in the PMS values in the MT region. But in the LMT region, only the PMS values of combination of the three wavelength ranges were located in the vertical region of LMT. The PMS value for all profiles averaged in the LMT region was 783 hPa and 808 hPa for the UV+TIR and UV+TIR+MW measurements, respectively. The PMS value was increased by approximately 3% by adding the MW measurement to the UV+TIR measurements, although the PMS value of the MW measurement itself was lower than 300 hPa in the LMT region." was added.

Pages 21 – 22 Figures 3 and 4

[Figure]

Figure 3. Values of DFS and PCE in the upper troposphere (UT, 215-383 hPa), middle troposphere (MT, 383-749 hPa), and lowermost troposphere (LMT, >749 hPa): (red) June 2009 in CEC, (purple) June 2009 in ECS, (green) December 2009 in CEC, (blue) December 2009 in ECS and (black) all of 20 profiles.

→

[Figure]

Figure 3. Values of DFS in the upper troposphere (UT, 215-383 hPa), middle troposphere (MT, 383-749 hPa), and lowermost troposphere (LMT, >749 hPa): (red) June 2009 in CEC, (purple) June 2009 in ECS, (green) December 2009 in CEC, (blue) December 2009 in ECS and (black) all of 20 profiles.

[Figure]

Figure 4. Same as Fig. 3 but for the PMS (left) and RRE (right). The gray shaded area represents the vertical region that corresponds to UT, MT and LMT.

Pages 23 – 24 Figures 5 and 6

The figure numbers of the averaging kernel plots were renumbered to Figures 5 and 6.

**GC 2-5**

Additional terms for error estimations: for such a new multispectral retrieval, I strongly recommend to estimate the contribution of additional terms of errors such as cross errors from joint fit of surface albedo and emissivity and other systematic errors (atmospheric and surface temperature, water vapor, etc).

**Answer to GC 2-5**

We deeply appreciate the referee for pointing it out. The main purpose of this feasibility study is evaluation of adding the MW limb measurement to the synergetic retrieval of the tropospheric ozone. We added the description of large error sources in the MW limb measurement of spectroscopic parameters. Time difference between the MW limb measurement and UV and TIR nadir measurements was also discussed as follows.

**Revisions to GC 2-5**

Page 12 Lines 16 – 30

"In this study, we showed that an introduction of the MW limb measurement had a certain effect

to increase the sensitivity of the tropospheric ozone retrieval. However, following issues might cause bias and uncertainties in the retrieval results and should be considered to implement this retrieval method to real measurements. Discrepancy in spectroscopic parameters for several wavelength ranges is one of the most important error sources. For the ozone retrieval using the MW limb measurement, spectroscopic parameters are the largest error sources. It was reported that approximately 3-5% error was caused by uncertainties in air-broadening coefficient and line intensity in the case of the SMILES observation (Kasai et al., 2013). It is comparable to the approximately 4% uncertainty in the spectroscopic parameters in the UV and TIR wavelength ranges (Gratien et al., 2010). The tangent height correction can also be a large error source for the MW limb measurement (Kasai et al., 2013). The tangent height is a key parameter to determine the field-of-view, thus uncertainty in the tangent height causes discrepancy of the atmospheric layer assumed in the simulation and the true atmospheric layer. This discrepancy critically affects to the retrieval of the ozone amount in both the stratosphere and the troposphere, and also might cause bias for correction of time delay between the MW limb measurement and the other nadir measurements. In this study, we assumed instruments onboard the ISS (low orbit) and the time difference of approximately five minutes could be ignored. If the time difference was long and its correction was required, three-dimensional atmospheric modeling should be performed including the field-of-view of the MW limb measurement." was added.

**GC 2-6**

A better description of the atmospheric scenario is needed. How model and airborne data are combined? How the 20 cases are chosen? What is the main characteristic (time of the day, region, ozone load, surface and atmospheric properties) of each scenario? These aspects should be explicitly commented and described in the text of the paper.

**Answer to GC 2-6**

We deeply appreciate this suggestion. We synthesized the vertical profiles of ozone, water vapor and temperature from three atmospheric profiles; a one-way nested global-regional air quality forecasting (AQF) system [Takigawa et al., 2007, 2009], the Modern Era Retrospective-Analysis for Research and Applications (MERRA) data [Rienecker et al., 2011] and the COSPAR International Reference Atmosphere (CIRA) [Fleming et al., 1990]. The aircraft observation data was used in the MERRA data, but we did not use the aircraft observation data directly. The statement "We made a total of 20 atmospheric scenarios from the model calculation and air-born observations made over two Asian areas (CEC and ECS) in June and December 2009. (Page 4 Lines 32 – 33)" was quite misleading, and we improved this statement as follows.

The reason why we chose the 20 scenarios over Asian area, CEC and ECS, was that China is one of the largest air-polluted countries in the world and the prediction of transboundary pollution is one of the main purposes of the Japanese future missions. The CEC, located between Beijing and Shanghai, is the area where largest amount of ozone in the lowermost troposphere was observed over East Asia from the Aura/OMI measurement [Hayashida et al., 2015]. This work also showed the amount of ozone in the lowermost troposphere in the urban area was largest in June (summer season) and smallest in December (winter season). The ECS was selected for a comparison between the urban area (CEC) and ocean (ECS). The main characteristics of the 20 scenarios are shown in Table 2, and we added the explanation of the 20 scenarios in our manuscript as follows.

[revised manuscript text omitted]

**GC 2-7**

Spectroscopy coherence should be mentioned as an important issue for real retrievals. There exist previous multispectral retrievals combining microwave spectra with other domains? Spectroscopic consistency has been analyzed as for UV and IR (e.g. Gratien et al., 2010)?

**Answer to GC 2-7**

We thank the referee for drawing our attention to this point. As far as we know, our study is the first attempt to introduce the MW measurement to the synergetic retrieval of the tropospheric ozone. We referred Gratien et al., (2010) and discussed the tropospheric ozone retrieval error due to uncertainties in the spectroscopic parameters in the MW ranges as follows.

"from 1.00 in the TIR+UV measurements to 1.23 (about a 20% increase)"

→

"from 1.00 to 1.23 (about 23% increase) for the TIR+UV measurements"

Page 11 Line 1

"in the MT and LT" → "in the UT and MT"

Page 11 Line 5

"a 40%" → "30%"

Page 12 Line 2

[revised manuscript text omitted]

[a] DFS values over land and ocean are averaged.

---

## Author Response (AR2)

**Comments from Referee**

For final publication, the manuscript should be: accepted subject to minor revisions

Suggestions for revision or reasons for rejection (will be published if the paper is accepted for final publication):

In the revised manuscript, Kasai et al. have followed or attempt to follow my previous recommendations. Thus the quality of the paper has improved.

In the two major corrections, two aspect should be better clarified (presentation quality will be good once this is done):

1) The following statement is not clear : "The SNR values were estimated by an interpolation with the small, middle and large values of the radiance calculated in the conditions that the surface albedo and solar zenith angle (SZA) of 90% and 0o, 25% and 45o, and 5% and 80o, respectively [Private communication with K. Gerilowski]" page 4, lines 25-27. I recommend to rewrite this statement and include in the new manuscript the Figure GC2.1.1 (or at least partially) which is very clear. If the figure can not be published in the manuscript by Kasai et al., at least a table recapitulating the values of SNR for the different values of SZA and surface albedo should be reported.

2) Simulations still provide retrieval error around 65%. In the revise manuscript, the calculation of a relative reduction of retrieval error is a better way to present it. However, we do not know if we obtain the same relative reduction of retrieval error by adding additional channels if the retrieval error itself is 65% or 20-30% as it is for real retrievals. I strongly recommend to do at least a calculation to prove that relative reduction of the retrieval error is the same in both situations (65% and 20-30% retrieval errors) and present it explicitly (with a table or figure) in the new manuscript.

These two remarks should be followed before the acceptance of the manuscript for publication.

**Reply to Referee**

Dear Referee

We would like to thank the Referee for the valuable comments and suggestions. Please find the manuscript with our revision. We answered to your comments and suggestions point by point. We performed a sensitivity study for degree of freedom for signal (DFS), pressure of maximum sensitivity (PMS) and relative reduction rate of error (RRE) with several a priori setting ($\sigma_a =$ 100%, 50%, 30%, 20%, 10%). We confirmed that the relative differences of DFS PMS and RRE for the wavelength combinations were the same for all $\sigma_a$ cases and that the retrieval sensitivity of the LMT ozone was increased by adding the MW measurement in all $\sigma_a$ cases. We hope that our manuscript is significantly improved. We still wonder our English is not perfect as a manuscript of AMT. We will ask the native English check again before submission of the final version.

Sincerely yours,

Tomohiro Sato and Yasuko Kasai
National Institute of Information and Communications Technology

**Suggestion 1**

1) The following statement is not clear : "The SNR values were estimated by an interpolation with the small, middle and large values of the radiance calculated in the conditions that the surface albedo and solar zenith angle (SZA) of 90% and 0o, 25% and 45o, and 5% and 80o, respectively [Private communication with K. Gerilowski]" page 4, lines 25-27. I recommend to rewrite this statement and include in the new manuscript the Figure GC2.1.1 (or at least partially) which is very clear. If the figure can not be published in the manuscript by Kasai et al., at least a table recapitulating the values of SNR for the different values of SZA and surface albedo should be reported.

**Answer**

We would like to thank you for the helpful suggestions. In the UV simulation, we prepared three typical SNR values for high, middle and low level of the radiance; then we linearly interpolated SNR values for the simulated radiance by using the two of three SNR values. As you mentioned, the summary table of the SNR values and the corresponding values of SZA and surface albedo used in the calculation of the radiances is a clear way to describe our calculation for SNR. We improved the statement and added the table in our manuscript as follows.

**Revision**

Page 4 Lines 24 – 29

"The SNR values were estimated by an interpolation with the small, middle and large values of the radiance calculated in the conditions that the surface albedo and solar zenith angle (SZA) of 90% and 0º, 25% and 45º, and 5% and 80º, respectively [Private communication with K. Gerilowski]. The mean values of SNR were estimated to be approximately 90 and 1400 at 305 nm and 340 nm, respectively."

$\rightarrow$

"We used SNR values of the APOLLO instrument setups in the UV simulation. Three references of the SNR value were prepared for high, middle and low level of the radiance. The SNR value for the simulated radiance was linearly interpolated by the two of the three reference SNR values. Table 2 shows the reference SNR values at 305 nm and 340 nm, and the solar zenith angle (SZA) and surface albedo of the three conditions [Private communication with K. Gerilowski]. The values of SNR used in the UV simulation were estimated to be approximately 90 and 1400 at 305 nm and 340 nm, respectively."

Page 27 Table 2 was added.

Table 2. Estimation of three reference SNR value in the UV simulation.

|  | SNR at 305 nm | SNR at 340 nm |
|---|---|---|
| Case 1: high level radiance (albedo 90 %, SZA 0°) | ~200 | ~2550 |
| Case 2: middle level radiance (albedo 25 %, SZA 45°) | ~60 | ~1200 |
| Case 3: low level radiance (albedo 5 %, SZA 80°) | ~10 | ~450 |

**Suggestion 2**

2) Simulations still provide retrieval error around 65%. In the revise manuscript, the calculation of a relative reduction of retrieval error is a better way to present it. However, we do not know if we obtain the same relative reduction of retrieval error by adding additional channels if the retrieval error itself is 65% or 20-30% as it is for real retrievals. I strongly recommend to do at least a calculation to prove that relative reduction of the retrieval error is the same in both situations (65% and 20-30% retrieval errors) and present it explicitly (with a table or figure) in the new manuscript.

**Answer**

We deeply appreciate your valuable comment. We agree your suggestion whether or not the same value of reduction rate of error (RRE) can be achieved if the level of retrieval error was different. The setting of a priori error ($\sigma_a$) is most sensitive for the retrieval error in our simulation, thus we, again, performed the calculations of the degree of freedom for signal (DFS), pressure of maximum sensitivity (PMS) and RRE using several conditions for $\sigma_a$ values ($\sigma_a$ = 100%, 50%, 30%, 20% and 10% of the log-based volume mixing ratios of a priori). In addition, we fixed a small bug in the previous calculation of RRE. The results of the new calculation were included in our manuscript as follows.

We also improved the definition of RRE. The RRE should represent the relative error reduction from the a priori error, thus we defined the RRE as

$$RRE = \frac{PCE_{apriori} - PCE_{retrieved}}{PCE_{apriori}},$$

While previously calculated as

$$RRE = \frac{PCE_{apriori} - PCE_{retrieved}}{PC}.$$

Here, where PCE$_{apriori}$ is partial column error (PCE) for the a priori state, PCE$_{retrieved}$ is PCE for the retrieved state, and PC is the partial column of ozone. Figure 4 was updated due to the re-definition of RRE as follows.

**Revision**

Page 10 Line 2

"$RRE = \frac{PCE_{apriori} - PCE_{retrieved}}{PC}$ [%]     (14)"

$\rightarrow$

"$RRE = \frac{PCE_{apriori} - PCE_{retrieved}}{PCE_{apriori}}$ [%]     (14)"

Page 11 Line 25

"25 – 45%" → "0 – 20%"

Page 11 Line 27

"36%" → "9%", "39%" → "11%"

Page 11 Line 28

"3%" → "2%"

Page 12 Lines 16 – 22

[revised manuscript text omitted]

[a] DFS values over land and ocean are averaged.

---

## Author Response (AR3)

**Reply to Editor**

Dear Editor

We greatly appreciate your efforts to help us improve our manuscript. In this version, we have worked to improve the quality of language throughout the manuscript. We would like to ask you to change the order of authors considering their contribution.
We hope that our manuscript is suitable for publication in AMT.

Sincerely yours,

Tomohiro Sato and Yasuko Kasai
National Institute of Information and Communications Technology

**Corrections by ourselves**

Page 1 Authors
The first author: Yasuko Kasai → Tomohiro O. Sato
The last author: Sachiko Hayashida → Yasuko Kasai

Page 1 Line 1
"to" → "of"
"the amount of ozone in the troposphere" → "the tropospheric ozone"

Page 1 Line 3
"sensitivity of the tropospheric retrieval" → "retrieval sensitivity of the tropospheric ozone"

Page 1 Lines 4 – 5
"The urban and sea areas in East Asia in summer and winter seasons were selected for the feasibility study." → "Two observation points in East Asia (one in an urban area and one in an ocean area) and two observation times (one during summer and one during winter) were assumed."

Page 1 Line 6
"the" was added.

Page 1 Line 7
"sensitivities of retrieved ozone" → "retrieval sensitivities of the ozone profiles"

Page 1 Line 8
"values of" was removed.
"the" was added.

Page 1 Lines 8 – 9
"error reduction rate" → "the reduction rate of error"

Page 1 Line 9
"the" was added.

Page 1 Lines 9 – 10
"The measurement noises were assumed at the same level as the currently available instruments."

→ "The measurement noise levels were assumed to be the same as those for currently available instruments."

Page 1 Line 14
"We found that the" → "The"

Page 1 Line 16
"of" → "value for"
"It might" → "These findings"

Page 1 Line 17
"of the" → "on"
"educed" → "obtained"

Page 1 Line 18
"will be implemented in the Japanese air-quality monitoring missions, APOLLO, GMAP-Asia and uvSCOPE" → "are applicable to the upcoming air-quality monitoring missions, APOLLO, GMAP-Asia and uvSCOPE"

Page 2 Line 3
"largest single environmental health risk" → "greatest environmental health risks"

Page 2 Lines 3 – 4
"Ozone in particular causes serious damage for human health and agricultural crops." → "Ozone in particular adversely affects human health and agricultural production."

Page 2 Line 8
"status and to make forecasts of future ozone amount" → "situation and to forecast future ozone amounts"

Page 2 Line 9
"surface ozone" → "ozone at the surface level"

Page 2 Line 10
"impact on the health of humans" → "effect on the health of people"
"is responsible for significant reduction in crop yields" → "that can significantly reduce crop

yields"

Page 2 Line 11
"so that" → "so"

Page 2 Line 15
"in the" was added.

Page 2 Line 16
"a variety of" → "various"

Page 2 Line 33
"ozone retrieval" → "ozone profile retrieval"

Page 3 Line 1
"on" → "using"
"the retrieval sensitivity of" → "retrieval sensitivity for"

Page 3 Line 2
"the" was added.

Page 3 Line 3
"the UV and TIR spectra of the OMI and TES measurements" → "UV and TIR spectra measured with the OMI and TES instruments"
"A value of the" → "The"

Page 3 Lines 4 – 5
"among OMI, TES, and ozonesonde" → "made using the OMI, the TES, and an ozonesonde"

Page 3 Line 7
"on" → "for"

Page 3 Line 8
", respectively" was removed.

Page 3 Line 9

"approximately 50% increased" → "increased by approximately 50%"

Page 3 Line 10
"only" → "alone"

Page 3 Line 11
"in" → "for"

Page 3 Lines 11 – 12
"They also showed a significant availability" → "The results demonstrated the effectiveness"

Page 3 Line 13
"improve" was added.
"profile" was added.

Page 3 Line 17
"can" → "should"
"missions for air-pollution" → "air-quality monitoring missions"

Page 3 Line 19 – 20
"APOLLO and GMAP-Asia mission" → "the APOLLO and GMAP-Asia missions"

Page 3 Line 21
"The missions of APOLLO and GMAP-Asia" → "These missions"

Page 3 Line 24
"the" was added.
"aid in the" was added.

Page 3 Line 25
"data-seta" → "datasets"

Page 3 Line 26
"data" → "datasets"

Page 3 Line 27

"ozone retrieval" → "ozone profile retrieval"

Page 3 Lines 28 – 29
"to obtain vertically resolved information of ozone within the troposphere not only at the boundary layer but also in the middle and upper troposphere by utilizing synergetic observation afforded by UV, TIR and MW instruments" → "utilizing synergetic observation afforded by UV, TIR, and MW instruments to obtain vertically resolved information on tropospheric ozone not only at the boundary layer but also in the middle and upper troposphere"

Page 3 Line 30
"ozone retrieval adding" → "ozone profile retrieval to the addition of"

Page 3 Lines 31 – 32
"Thus, the feasibility study was performed under an ideal condition for the synergetic retrieval of the tropospheric ozone." → "The simulation was thus performed under ideal conditions for synergetic retrieval of the tropospheric ozone profile."

Page 4 Line 3
"used in the simulation" was added.

Page 4 Lines 3 – 4
"We assumed three spectrometers equipped in ISS that" → "Three spectrometers in the ISS were assumed to"

Page 4 Line 4
"of" → ":"

Page 4 Line 5
"use" → "used"
"measurement uses" → "one used"

Page 4 Line 7
"the" was added.
"in our simulation" was removed.

Page 4 Line 8

"set to parallel" → "set parallel"

Page 4 Lines 10 – 11
"transport when we assume a typical value of horizontal wind speed in the troposphere and the stratosphere of 1.2 km/min" → "travel if a typical value of horizontal wind speed in the troposphere and the stratosphere (1.2 km/min) is assumed"

Page 4 Line 11
"smaller than" → "less than the"

Page 4 Line 13 – 14
"Table 1 is a summary of the specification of the three assumed instruments and the radiative transfer models used in this study." → "Table 1 summarizes the specifications of the three instruments and the three radiative transfer models."

Page 4 Line 15
"of the simulation of" → "for"
"The UV wavelength ranges" → "UV wavelength range"

Page 4 Line 16
"ozone retrieval" → "ozone profile retrieval"
"In our simulation, we" → "We"

Page 4 Line 17
"for the" → "to"

Page 4 Lines 17 – 20
"the stratospheric ozone retrieval by the MW measurement improves the tropospheric ozone retrieval sensitivity. We also decided not to include the VIS (340-505 nm) range in this study, although the benefit of adding VIS wavelengths has been reported (e.g., Sellitto et al., 2012, b). The reason why we excluded these ranges is because the wavelength dependence of the surface reflectance, absorption of $NO_2$ and the Ring effect were out of the scope of the study." → "stratospheric ozone profile retrieval using MW measurement improves the sensitivity of tropospheric ozone profile retrieval. Although there is a benefit of adding VIS wavelengths (340-505 nm) (e.g., Sellitto et al., 2012, b), we decided not to because the wavelength dependence of surface reflectance, the absorption of $NO_2$, and the Ring effect were beyond the scope of this

study.”

Page 4 Line 28
“would be” → “was”

Page 4 Line 32
“in” → “over”

Page 5 Lines 1 – 2
“The MW limb-sounding instrument considered in this study was designed for covering two frequency bands, i.e., the 350 GHz band (345-357 GHz) and 645 GHz band (639-651 GHz).” → “The assumed MW limb-sounding instrument covered two frequency bands: the 350 GHz band (345-357 GHz) and the 645 GHz band (639-651 GHz).”

Page 5 Line 8
“We also assumed that a” → “We assumed that the”

Page 5 Line 9
“that” was added.

Page 5 Line 11
“350 GHz band and 645 GHz band” → “the 350 GHz and 645 GHz bands”

Page 5 Lines 13 – 14
“We performed a feasibility study of tropospheric ozone retrieval for typical atmospheric scenarios in summer and winter. The target area of this study is East Asia, one of most serious ozone polluted areas and the source of ozone intercontinental transport toward North America from Asia.” → “We used typical atmospheric scenarios in summer and winter for East Asia, which is one of most ozone-polluted regions and a major source of the intercontinental transport of ozone toward North America.”

Page 5 Line 16
“, centered among China, Japan, South Korea, and Taiwan” was added.

Page 5 Lines 21 – 22
“it was shown that the ozone retrieval sensitivity was most increased in small to moderate SZAs

in the simulation study performed by Landgraf and Hasekamp (2007)" → "ozone profile retrieval sensitivity is apparently higher when the solar zenith angle (SZA) is low or moderate, as shown in a simulation study by Landgraf and Hasekamp (2007)"

Page 5 Line 23
"over the two Asian areas (CEC and ECS) in June and December 2009. The characteristics of the 20 atmospheric scenarios are presented" → ", as shown"
", and the" → ". The"

Page 5 Lines 24 – 26
"We discuss the simulation results, dividing into four cases (CEC in June, ECS in June, CEC in December and ECS in December). The ozone partial column (PC) in LMT at CEC in June was largest (approximately $5\times10^{21}$ m$^{-2}$) among those of four cases, and the smallest value (approximately $2\times10^{21}$ m$^{-2}$) was taken from the case at CEC in December." → "Of the four cases (CEC in June, ECS in June, CEC in December, and ECS in December), the ozone partial column (PC) in the LMT was the largest (approximately $5\times10^{21}$ m$^{-2}$) for CEC in June and was the smallest (approximately $2\times10^{21}$ m$^{-2}$) for CEC in December."

Page 5 Line 27
"of ozone, temperature and water vapor" was added.

Page 6 Line 4
"the" was added.

Page 6 Line 7
", and" → ";"

Page 6 Line 8
"to be equal to the value at the surface pressure," → "equal to the value at surface pressure"

Page 6 Line 10
"in a vertical region of 985-0.01 hPa" → "for the 985-0.01 hPa vertical range"

Page 6 Lines 11 – 12
"A data product named ``MERRA DAS 3d analyzed state (inst6_3d_ana_Nv)'' provided the three-dimensional fields of layer pressure thickness" → "The ``MERRA DAS 3d analyzed state

(inst6_3d_ana_Nv)" data product provided three-dimensional fields for layer pressure thickness"

Page 6 Line 14
"nearest" → "closest"

Page 6 Line 17
"model" was added.

Page 6 Line 18
"model" was added.

Page 6 Lines 21 – 22
"because there are no appropriate data to refer" → "due to the lack of appropriate reference data"

Page 6 Line 22
"in" → "for"

Page 6 Line 28
"that were described in" → "as described by"

Page 6 Line 29
". The" → " because the"

Page 6 Line 32
"selected UV ranges" → "target UV range"

Page 7 Line 2
"impact" → "effect"

Page 7 Line 3
"in" → "by"

Page 7 Line 5
"MW" → "the MW measurement"

Page 7 Lines 13 – 14

"In the presented study, no bias is assumed between the three forward models in order to investigate potential advantage of including MW observation to retrieval of tropospheric ozone." → "To investigate the potential advantage of including MW observations in the retrieval of the tropospheric ozone profile, we assumed no bias between the three forward models."

Page 7 Line 16
"Near Infrared" → "near infrared"

Page 7 Line 21
", and it" → "and"

Page 7 Line 29
"are" → "were"
"HITRAN 2008" → "the HITRAN 2008 molecular spectroscopic database"

Page 7 Line 31
"are also included and are" → "was taken into account; it is"

Page 8 Line 2
"system and their error estimations" → "and error estimation"

Page 8 Line 12
"which" → "that"
"summation of" → "summing the"

Page 8 Line 16
"influenced" → "affected"

Page 8 Line 22
"the" was added.

Page 8 Line 29
"due to the measurement" was removed.

Page 8 Line 29 – Page 9 Line 5
"The diagonal components of $S_\varepsilon$ were the squares of the measurement error $\sigma_\varepsilon$. The off-diagonal

components of $S_\varepsilon$ were set to zero. The off-diagonal components in $S_a$ indicate the correlations between the ozone concentrations in different vertical layers. Non-zero off-diagonal components in $S_a$ assist the retrieval of ozone concentration in a layer in which sufficient ozone information is not included in a measurement spectrum with the correlations with other layers in which sufficient ozone information is included in a measurement spectrum (e.g., Saitoh et al., 2009). One of the aims of our feasibility study is to investigate the ozone retrieval sensitivity for each vertical region in an ideal condition. We set off-diagonal components in $S_a$ to be zero to avoid the assistance of the ozone retrieval by the correlations between different vertical layers." → "The off-diagonal components in $S_a$ indicate the correlations between the ozone concentrations in different vertical layers. The non-zero off-diagonal components in $S_a$ facilitate retrieval of the ozone concentration in a layer for which sufficient ozone information is not included in the measurement spectrum with the correlations with other layers in which sufficient ozone information is included in a measurement spectrum (e.g., Saitoh et al., 2009). We set the off-diagonal components in $S_a$ to zero to avoid the assistance of the ozone retrieval by the correlations between different vertical layers. The diagonal components of $S_\varepsilon$ were the squares of the measurement error $\sigma_\varepsilon$. The off-diagonal components of $S_\varepsilon$ were set to zero."

Page 9 Line 6
"with" → "of"

Page 9 Line 7
"in computational calculation" was removed.

Page 9 Line 10
"was" → "is"

Page 9 Line 12
"whose" → "in which the"

Page 9 Line 16
"Using the normalized vectors and matrices, A and S are expressed as" → "A and S are expressed using the normalized vectors and matrices as"

Page 9 Line 19, 23
"ozone retrieval" → "ozone profile retrieval"

Page 10 Line 4

"the" was added.

Page 10 Lines 6 – 8

"The sensitivity of ozone retrieval for the UT (215-383 hPa), MT (383-749 hPa), and LMT (>749 hPa) regions was investigated in terms of DFS. Figure 3 and Table 4 show the DFS values calculated with Eq. (13) for the seven wavelength combinations: UV alone, TIR alone, MW alone, TIR+MW, UV+MW, UV+TIR and UV+TIR+MW. The DFS values were averaged in June in CEC (shown by red markers in Fig. 3), June in ECS (purple), December in CEC (green), December in ECS (blue) and all 20 profiles (black). The error bar represents the standard deviation." → "The sensitivities of ozone profile retrieval in terms of DFS (calculated using Eq. (13)) are plotted in Fig. 3 and summarized in Table 4. The average DFS values are plotted in red for CEC in June, in purple for ECS in June, in green for CEC in December, in blue for ECS in December, and in black for all 20 profiles. The error bars represent the standard deviation."

Page 10 Line 9

"value" → "values"

"was" → "were"

Page 10 Line 10

"DFS average" → "average DFS"

Page 10 Line 11

"Using more than one wavelength range" → "For multiple-wavelength measurement"

"value" → "values"

Page 10 Line 12

"for the wavelength combinations of TIR+MW, UV+MW, UV+TIR, and UV+TIR+MW" → "for the TIR+MW, UV+MW, UV+TIR, and UV+TIR+MW measurement combinations"

Page 10 Line 13

"of" → "for"

Page 10 Lines 13 – 15

"those of more than one wavelength range, and adding the MW region increased the value by about two times. The additional MW region was hence most effective at improving the retrieval

of ozone in the UT region." → "the four multiple-wavelength measurements; adding the MW measurement approximately doubled the DFS value. The addition of the MW measurement was thus the most effective way of improving the retrieval of the ozone profile in the UT region."

Page 10 Line 16
"measurements are the main contributors" → "measurement was the main contributor"

Page 10 Line 18
"of" → "for"

Page 10 Lines 20 – 22
"the MW measurements, which have no information on ozone in the MT region because of atmospheric opacity, certainly increased the DFS value in the MT region from 1.00 to 1.23 (about 23% increase) for the TIR+UV measurements" → ", although the MW measurement provided no information on ozone in the MT region because of atmospheric opacity, it nevertheless increased the DFS in the MT calculation from 1.00 to 1.23 (approximately 23%) for the TIR+UV measurement"

Page 10 Line 22
"the" was removed.

Page 10 Line 23
"MW" → "the MW measurement"
"ozone in" → "the ozone profile for"

Page 10 Line 27
"that in" was added.

Page 10 Lines 27 – 28
"value of the UV+TIR measurements" → "value for the UV+TIR measurement"

Page 10 Line 28
"increase" was removed.

Page 10 Line 29
"in the case at" → "for"

"2009" was removed.

"the" was added.

Page 10 Line 30

"of" → "in"

Page 10 Lines 30 – 32

"In general, the discontinuity in the averaging kernel occurs because of mathematical issues not atmospheric physical issues, thus we avoid any scientific discussion with the DFS value for the TIR measurement in this case (ECS in December 2009)." → "Since discontinuity generally occurs in the averaging kernel due to mathematical factors, not atmospheric physical factors, we do not discuss the DFS value for the TIR measurement in the ECS in December case."

Page 11 Line 1X

"the" → "those of"

Page 11 Line 2

"were" → "are"

Page 11 Lines 2 – 3

"Scenarios of the simulation or the measurements are different among this work and the previous studies shown in Table 5, thus, the DFS values themselves should not be directly compared each other." → "Since the scenarios of the simulation or measurement are different among this work and the previous studies, the DFS values cannot be directly compared."

Page 11 Line 3

"Here we" → "We thus"

Page 11 Line 4

"difference" → "differences"

"value" → "values"

"the" was added.

Page 11 Line 5

"the" was added.

"of" → "for"

Page 11 Lines 5 – 6

"estimated to be 126%. It showed good agreement with" → "126%, which shows good agreement with"

Page 11 Lines 7 – 8

"in a range of the corresponding partial column" → "within the corresponding vertical region"

Page 11 Lines 8 – 13

"In the UT region, the PMS values for all cases were located in a range of the corresponding region (215-383 hPa) by combining more than two wavelength ranges. It was also observed in the PMS values in the MT region. But in the LMT region, only the PMS values of combination of the three wavelength ranges were located in the vertical region of LMT. The PMS value for all profiles averaged in the LMT region was 783 hPa and 808 hPa for the UV+TIR and UV+TIR+MW measurements, respectively. The PMS value was increased by approximately 3% by adding the MW measurement to the UV+TIR measurements, although the PMS value of the MW measurement itself was lower than 300 hPa in the LMT region." → "The PMS values for the UT calculation for all cases were located within the corresponding range (215-383 hPa) when two or more wavelength ranges were combined. This was also observed for the MT calculation. For the LMT calculation, this was observed only when three wavelength ranges were combined. The PMS values for all profiles averaged for the LMT were 783 hPa and 808 hPa for the UV+TIR and UV+TIR+MW measurements, respectively. The PMS value was increased approximately 3% by adding the MW measurement to the combined UV+TIR measurement although the PMS value in the MW measurement for the LMT was lower than 300 hPa."

Page 11 Line 14

"in the ozone partial column of ozone" → "for the partial column of ozone"

", is shown in the right column of" → "is shown on the right in"

Page 11 Line 15

"value of" was removed.

Page 11 Lines 15 – 19

"The RRE generally increased by combining more wavelength ranges in the ozone synergetic retrieval as DFS shown in Fig. 3. The RRE value for all profiles averaged in the LMT region was 9% and 11% for UV+TIR and UV+TIR+MW measurements, respectively. Adding the MW

measurement made 2% increase of RRE value. A certain increase of the retrieval sensitivity of the LMT ozone was shown by DFS as well as PMS and RRE." → "It was generally higher when more wavelength ranges were combined in the ozone synergetic retrieval, as shown by the DFS values plotted in Fig. 3. The RRE value for all profiles averaged was 36% for the LMT calculation and 39% for the UV+TIR and UV+TIR+MW measurements. Adding the MW measurement increased the RRE by 3%. The DFS, PMS, and RRE values all show a certain increase in the retrieval sensitivity of the LMT ozone profile."

Page 11 Line 20
"in this simulation" was removed.

Page 11 Lines 20 – 21
"its dependency transferred to" → "this dependency was reflected in"

Page 11 Lines 21 – 22
"The DFS values of the MW measurement in the UT region for profiles in December 2009 (green and blue markers in Fig. 3) were larger than those in June 2009 (red and purple markers)." → "As shown in Fig. 3, the DFS values in the MW measurement for the UT in December for both areas were larger than those for June."

Page 11 Line 23
"ozone in the UT region" → "the UT ozone"

Page 11 Line 24
"strongly depended" → "depended greatly"

Page 11 Line 25
"at CEC in June 2009" → "for CEC in June"

Page 11 Lines 25 – 26
"of UV+TIR+MW at CEC in June 2009 (red marker)" → "in the UV+TIR+MW for CEC in June"

Page 11 Line 27
"with" → "using the averaging kernel matrix"

Page 11 Line 28

"simulation using the atmospheric profile ¥#01 and ¥#12 for all of the" → "simulations using atmospheric profiles #01 and #12 for all"

Page 11 Line 29
"i.e.," was removed.

Page 11 Line 30
"In the case of" → "For"

Page 11 Line 31
"in" → "on"
"the" was added.

Page 11 Line 33, 34
"of" → "for"

Page 11 Lines 34 – 35
"as a result of adding the two measurements" → "when the two measurements were combined"

Page 11 Line 35
"The combination of TIR and UV improved" → "Combination of these measurements improves"

Page 11 Line 35 – Page 12 Line 1
"ozone in the LMT" → "the LMT ozone amount"

Page 12 Line 2
"In" → "For"

Page 12 Line 4
"adding" → "combining"

Page 12 Line 7
"of" was removed.

Page 12 Line 10
"It was shown that the increase" → "Increase"

"amount" was added.

Page 12 Lines 12 – 14
"In this study, we showed that an introduction of the MW limb measurement had a certain effect to increase the sensitivity of the tropospheric ozone retrieval. However, following issues might cause bias and uncertainties in the retrieval results and should be considered to implement this retrieval method to real measurements." → "Our results show that introducing MW limb measurement increases the sensitivity of tropospheric ozone profile retrieval. However, several factors may cause bias and uncertainty in the retrieval results and thus should be considered before the proposed retrieval method is implemented."

Page 12 Line 14
"the" was added.

Page 12 Lines 14 – 15
"one of the most important error sources" → "a major source of errors"

Page 12 Lines 15 – 16
"For the ozone retrieval using the MW limb measurement, spectroscopic parameters are the largest error sources" → "For ozone profile retrieval using MW limb measurement, the spectroscopic parameters are the major error source"

Page 12 Line 16
"approximately 3-5% error" → "an error of approximately 3-5%"
"the" was added.

Page 12 Line 17
". It" → ", which"

Page 12 Line 18
"The tangent height" → "Tangent height"

Page 12 Line 20
"thus" → "so"
"discrepancy of" → "a discrepancy between"

Page 12 Line 21
"to" was removed.

Page 12 Line 22
"amount" → "profile"
"also might" → "may"
"for correction of" → "in the correction of the"

Page 12 Line 22, 23
"the" was removed.

Page 12 Line 23
"that the" was added.
"Earth" was added.

Page 12 Line 24
"the" → "that a"

Page 12 Lines 24 – 26
"If the time difference was long and its correction was required, three-dimensional atmospheric modeling should be performed including the field-of-view of the MW limb measurement." → "If the time difference is actually long enough to need correction, three-dimensional atmospheric modeling including the field-of-view of the MW limb measurement should be performed."

Page 12 Line 27
"profile" was added
  "the" was removed.

Page 12 Line 28
"to increase about" → "to be increased by approximately"

Page 12 Line 29
"value was" → "values were"

Page 12 Lines 29 – 30
"future mission of IASI-NG and UVNS" → "upcoming IASI-NG and UVNS missions"

Page 12 Line 30

"the" was removed.

"this" → "the"

Page 12 Lines 31 – 32

"showed a possibility to retrieve the ozone" → "has shown that it is possible to retrieve the ozone profile"

Page 13 Line 2

"a vertically resolved ozone profile in the troposphere from" → "vertically resolved ozone profiles in the troposphere with"

Page 13 Line 3

"a combination of three separate" → "various combinations of three"

"Observation" → "The observation"

Page 13 Lines 4 – 5

"low orbit at a height of 300 km (the height of the ISS)" → "low Earth orbit (300 km, the height of the ISS)"

Page 13 Line 6

"retrieval of ozone in" → "tropospheric ozone profile retrieval for"

Page 13 Line 8

"," → "and"

Page 13 Line 10

"ozone" → "the ozone profile"

Page 13 Line 11

"additional" → "addition of"

Page 13 Line 11

"the" was removed.

Page 13 Line 12

"of" was removed.

Page 13 Line 15

"measurements" → "measurement"

Page 13 Line 16

"ozone" → "the ozone profile"

"strongly" → "greatly"

Page 13 Lines 16 – 17

"The DFS value in the LMT became larger in the case of" → "It was larger for a"

Page 13 Lines 17 – 18

"the UV+TIR+MW measurement of 1.03±0.01 was given by the case at" → "UV+TIR+MW measurement (1.03±0.01) was obtained for"

Page 13 Line 18

"2009" was removed.

"of" → "with"

Page 13 Line 19

"of approximately $5\times10^{21}\text{m}^{-2}$" → "(approximately $5\times10^{21}\text{m}^{-2}$)"

Page 13 Line 20

"derived" → "provided"

Page 13 Line 21

"at" → "in a"

"regions" was removed.

Page 13 Lines 21 – 22

"measurements to the UV and TIR measurement combinations" → "measurement to the combined UV and TIR measurement"

Page 13 Line 23, 24

"measurements" → "measurement"

Page 13 Lines 24 – 25
"This might indicate that reducing the uncertainty of ozone abundance in the stratosphere is important for estimating an accurate tropospheric ozone profile." → "This indicates that reducing uncertainty about ozone abundance in the stratosphere may be important for accurately estimating the tropospheric ozone profile."

Page 18 Fig. 1
Caption, "Geometries of down-looking nadir (UV and TIR) and limb (MW) observations" → "Geometries used for observation: nadir (UV and TIR) and limb (MW)."

Page 19 Fig. 2
Caption, "The 20 atmospheric scenarios used in this study: (a) VMR of ozone, (b) temperature, and (c) VMR of water vapor. These atmospheric scenarios can be divided into four groups, as denoted by four different color curves: (red) June 2009 in CEC, (purple) June 2009 in ECS, (green) December 2009 in CEC, and (blue) December 2009 in ECS. Two example profiles #01 and #12, represented by the black and gray lines, respectively, are the ones used to obtain the results shown in Figs. 5 and 6." → "Twenty atmospheric scenarios used in this study: (a) VMR of ozone, (b) temperature, and (c) VMR of water vapor. They are divided into four groups, as denoted by color: CEC–June 2009 (red), ECS-June 2009 (purple), CEC–December 2009 (green), and ECS–December 2009 (blue). Two example profiles (#01 and #12, black and gray lines, respectively) were used to obtain results shown in Figs. 5 and 6."

Page 20 Fig. 3
Caption, "Values of DFS in the upper troposphere (UT, 215-383 hPa), middle troposphere (MT, 383-749 hPa), and lowermost troposphere (LMT, >749 hPa): (red) June 2009 in CEC, (purple) June 2009 in ECS, (green) December 2009 in CEC, (blue) December 2009 in ECS and (black) all of 20 profiles." → "Values of DFS for upper troposphere (UT, 215-383 hPa), middle troposphere (MT, 383-749 hPa), and lowermost troposphere (LMT, >749 hPa): CEC–June 2009 (red), ECS–June 2009 (purple), CEC–December 2009 (green), ECS–December 2009 (blue), and 20 profiles averaged (black)."

Page 21 Fig. 4
Caption, "Same as Fig. 3 but for the PMS (left) and RRE (right). The gray shaded area represents the vertical region that corresponds to UT, MT and LMT." → "Same as Fig. 3 but for PMS (left)

and RRE (right). Gray shaded area represents vertical region comprising UT, MT, and LMT."

Page 22 Fig. 5

[revised manuscript text omitted]

[a] DFS values over land and ocean are averaged.